# Polygenic risk score for type 2 diabetes shows context-dependent effects across populations

Polygenic risk scores hold prognostic value for identifying individuals at higher risk of type 2 diabetes. However, further characterization is needed to understand the generalizability of type 2 diabetes polygenic risk scores in diverse populations across various contexts. We systematically characterize a multi-ancestry type 2 diabetes polygenic risk score among 244,637 cases and 637,891 controls across diverse populations from the Population Architecture Genomics and Epidemiology Study and 13 additional biobanks and cohorts. Polygenic risk score performance is context dependent, with better performance in those who are younger, male, without hypertension, and not obese or overweight. Additionally, the polygenic risk score is associated with various diabetes-related cardiometabolic traits and type 2 diabetes complications, suggesting its utility for stratifying risk of complications and identifying shared genetic architecture between type 2 diabetes and other diseases. These findings highlight the need to account for context when evaluating polygenic risk score as a tool for type 2 diabetes risk prognostication and the potentially generalizable associations of type 2 diabetes polygenic risk score with diabetes-related traits, despite differential performance in type 2 diabetes prediction across diverse populations. Our study provides a comprehensive resource to characterize a type 2 diabetes polygenic risk score.

The rise in type 2 diabetes (T2D) prevalence is a major public health challenge that shows no signs of abating. Without sufficient efforts to combat this pandemic, it has been estimated that 11.3% of the global population will have diabetes by 2030[1], at which time its economic burden is projected to be 2.2% of the global gross domestic product[2]. These predictions provide an urgent incentive to improve diabetes risk prediction, stratification, and prevention.

An emerging and promising approach to quantify T2D risk is the use of polygenic risk scores (PRS), which model the aggregate effect of genetic variants found across the genome on a disease or trait of interest. Such scores, combined with non-genetic predictors, potentially hold utility for identifying individuals at high risk prior to disease onset and could guide positive health behavior changes, screening practices, or therapeutic interventions[3–5]. These PRS have been facilitated by genome-wide association studies (GWAS) that have identified genetic variants conferring T2D susceptibility, with several recent T2D GWAS and fine mapping studies conducted across diverse populations enabling the development of multi-ancestry T2D PRS[6–8].

Given the large number of risk factors for T2D, there is a clear need to further characterize PRS. Although previous studies have reported that interactions between individual genetic variants and certain lifestyle factors influence T2D risk in European ancestry populations[8–11], little is known about whether the predictive accuracy and utility of T2D PRS are context-dependent and modified by individual characteristics and metabolic parameters, particularly across diverse populations. Moreover, it remains unclear whether PRS for T2D risk could hold prognostic value for diabetes-related complications across numerous organ systems and whether such associations

✉ e-mail: bguo@fredhutch.org; bdarst@fredhutch.org

are consistent across populations. While a recent investigation found that a T2D PRS was associated with T2D-related retinopathy, with modest evidence of association with chronic kidney disease, peripheral artery disease, and neuropathy in individuals who were genetically similar to European ancestry[7], no study to date has systematically evaluated diabetes-related complications across diverse populations. A thorough investigation of these and other complications across diverse populations could provide biological insights into the pleiotropic nature and prognostic ability of T2D PRS.

The Population Architecture using Genomics and Epidemiology (PAGE) Study, which consists of multiple large and diverse population-based studies, is uniquely situated to address these questions. In PAGE, we previously found that a multi-ancestry T2D PRS outperformed population-specific T2D PRS and explained up to 15.3% of familial relative risk of T2D across populations[8]. Here, after identifying a multi-ancestry T2D PRS that was optimally predictive of T2D risk across our diverse PAGE populations, we investigated whether the effect of a T2D PRS was context-dependent across PAGE and several independent biobanks and cohorts. To further understand the clinical implications of genetic risk of T2D on other diseases and conditions, we examined whether a T2D PRS was associated with T2D-related phenotypic risk factors and

complications and additionally conducted phenome-wide association studies (PheWASs) across diverse populations. The goal of this investigation is to improve precision medicine in diabetes by systematically evaluating the performance and context-dependent effects of a T2D PRS across diverse populations. Our findings serve as a comprehensive resource to characterize T2D PRS.

## Results
### Participants
An overview of the study design is provided in Fig. 1. In total, 82,944 participants from PAGE and 835,241 participants from 13 additional biobanks and cohorts were included in this study: 244,637 T2D cases, 637,891 controls, and 35,657 individuals with prediabetes. This included individuals who self-reported as African, African American, or Black (AFR); American Indian, Alaskan Native, or Native American (AI); Asian American, East of Southeast Asian, or South Asian/Indian (ASN); White or European American (EUR); Greenlandic; Hispanic/Latino American (HIS); Middle Eastern, North African, or Qatari (MENAQ); and Native Hawaiian or Other Pacific Islander (NH) (Table 1, Supplementary Data 1 and 2, see the "Methods" section, Supplemental Information for population descriptor details).

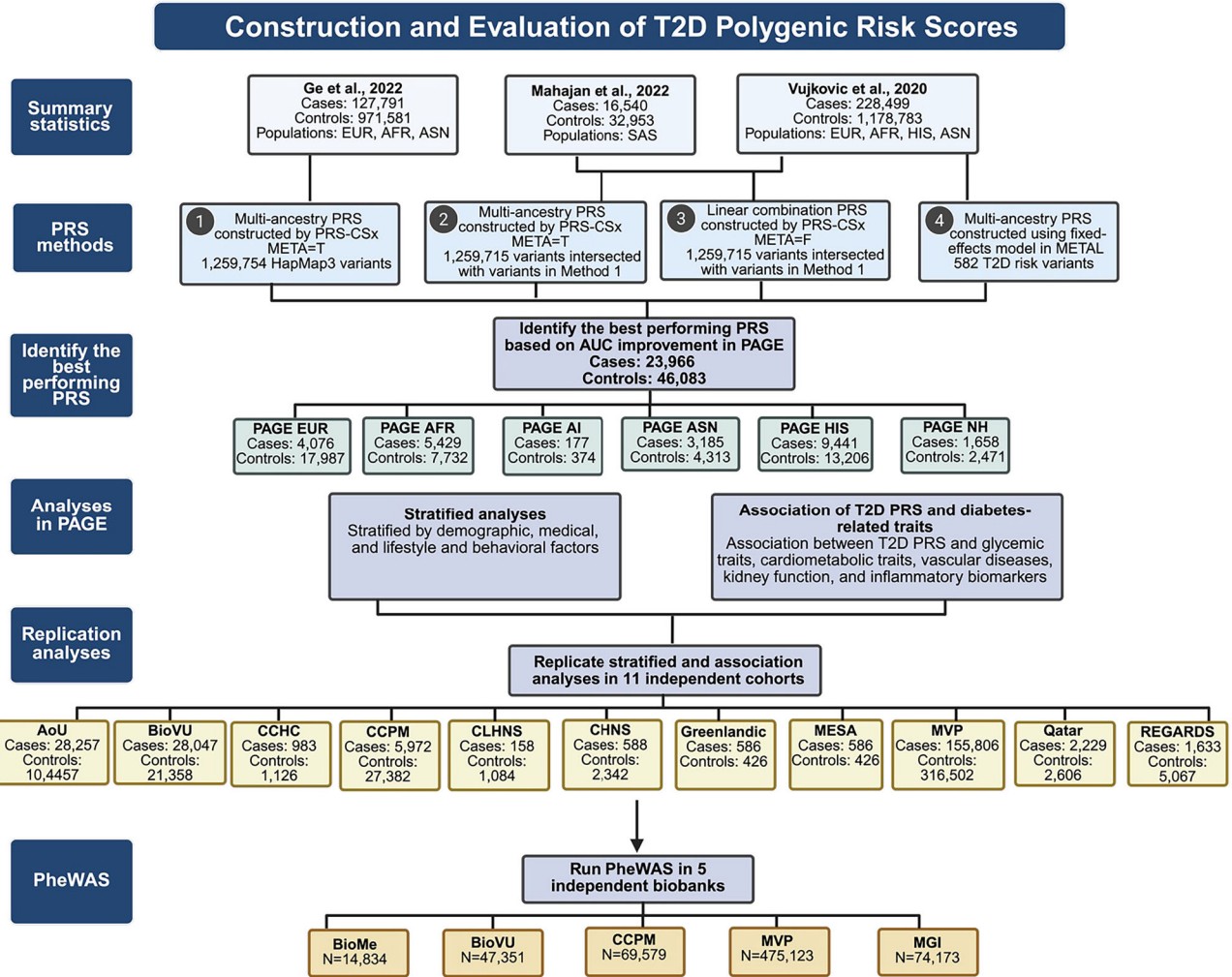

**Fig. 1 | Workflow of T2D PRS construction and evaluation.** Four T2D PRS were constructed and evaluated in the Population Architecture using Genomics and Epidemiology (PAGE) study to identify the best-performing score for downstream analyses. The best-performing PRS across populations was used for stratified analyses across demographic, medical, and lifestyle and behavioral characteristics, and association analyses with diabetes-related traits in PAGE. Stratified and association analyses were replicated in 11 independent cohorts. Phenome-wide association studies (PheWAS) were conducted across five independent biobanks to assess the broader clinical impact of the T2D PRS.

**Table 1 | Sample sizes of populations included by T2D status**

| Population | T2D cases | T2D controls | Prediabetes | Total |
|---|---|---|---|---|
| African, African American, or Black (AFR) | 52,159 | 106,306 | 2333 | 160,798 |
| American Indian, Alaskan Native, or Native American (AI) | 177 | 374 | 91 | 642 |
| Asian American, East of Southeast Asian, or South Asian/Indian (ASN) | 6999 | 17,660 | 3134 | 27,793 |
| White or European American (EUR) | 148,230 | 441,879 | 18,126 | 608,235 |
| Greenlandic | 586 | 426 | 2824 | 3836 |
| Hispanic/Latino American (HIS) | 32,440 | 65,613 | 8955 | 107,008 |
| Middle Eastern, North African, or Qatari (MENAQ) | 2352 | 3078 | 53 | 5483 |
| Native Hawaiian or Other Pacific Islander (NH) | 1694 | 2555 | 141 | 4390 |
| Total | 244,637 | 637,891 | 35,657 | 918,185 |

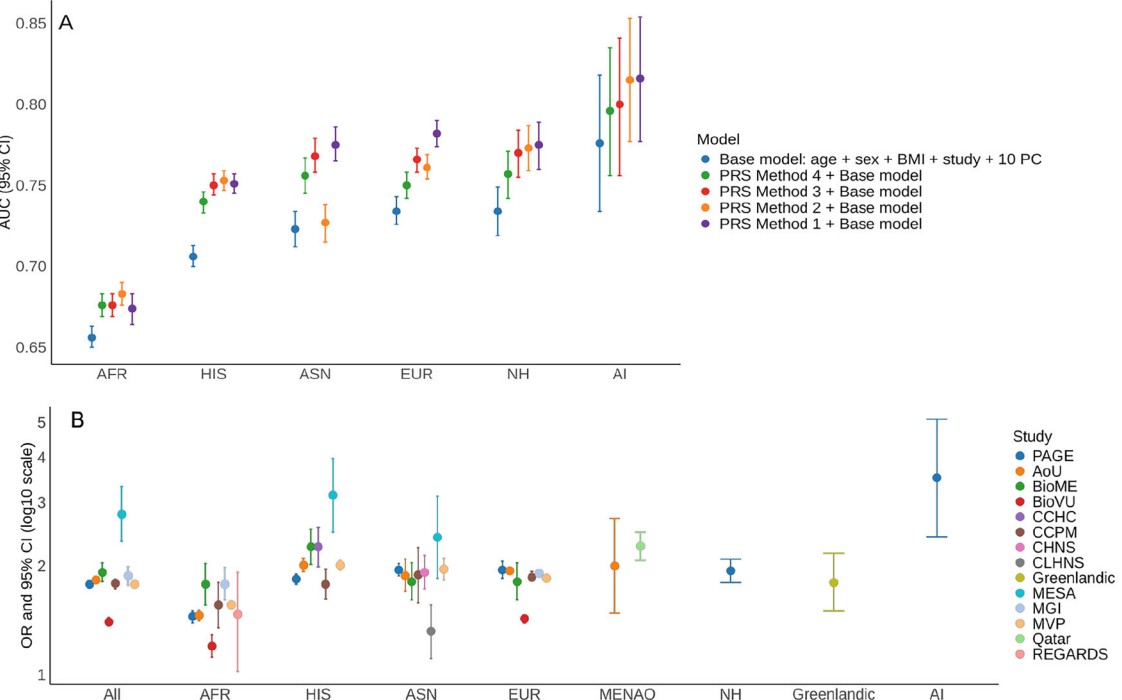

**Fig. 2 | Performance of T2D PRS across self-identified populations. A** Area under the curve (AUC) for four T2D PRS methods in PAGE. PRS Methods 1–4 correspond to those in Fig. 1. Data are presented as AUC with 95% confidence interval (CI). **B** Association between each SD unit increase in the Ge et al. T2D PRS with T2D risk in PAGE and the additional biobanks and cohorts. Data are presented as odds ratios (OR) with 95% CI. Sample sizes for **A** and **B** are provided in Supplementary Data 1.

## Evaluation of T2D PRS performance

Given our unique populations and how rapidly PRS are superseded, our first goal was to identify a T2D PRS that had optimal predictive ability across our populations to be carried forward in subsequent PRS evaluations. In PAGE, we assessed four distinct multi-ancestry PRS (Fig. 1, Supplementary Data 3, see the "Methods" section). The first three PRS were genome-wide PRS developed using 1.259 million Hap-Map3 variants in PRS-CSx[12], an approach that has demonstrated high predictive ability across several traits[13–15]. *PRS Method 1*: a previously developed T2D PRS[16] based on AFR[17], ASN[18], and EUR[19] GWAS summary statistics using meta-analyzed posterior effects with the META = T function in PRS-CSx; *PRS Method 2*: a new T2D PRS that we developed using a similar approach as PRS Method 1 but leveraging GWAS summary statistics from larger sample sizes of AFR, ASN, and EUR populations and also including HIS and South Asian (SAS) populations[6,7]; *PRS Method 3*: a new T2D PRS that we developed using the same summary statistics as PRS Method 2 but calculated from a linear combination of population-specific PRS using the META = F function in PRS-CSx; and *PRS Method 4*: a previously developed PRS with 582 genome-wide significant variants reported in AFR, ASN, EUR, and HIS

populations[7,8]. PRS derived from each method was standardized to the distribution of controls within each population.

Each of the four T2D PRS methods exhibited good predictive performance in PAGE (Supplementary Data 4). Although confidence intervals of the area under the receiver operating characteristic curve (AUC) overlapped between the methods, PRS Method 1 showed slightly better predictive performance across most populations compared to other methods. PRS Method 1 had AUC improvements of 1.8%, 4.0%, 5.2%, 4.8%, 4.5%, and 4.1% for AFR, AI, ASN, EUR, HIS, and NH populations, respectively, compared to the base model of age, sex, BMI, and the first 10 genetic principal components (PCs) (Fig. 2A and Supplementary Data 4). We used PRS Method 1 for all subsequent analyses.

In PAGE, the PRS was highly predictive of T2D risk across populations but with variable performance (Supplementary Fig. 1). Each standard deviation (SD) unit increase in PRS was associated with 1.78-fold (95% CI: 1.75, 1.82) higher odds of T2D across all populations combined, ranging from 1.45 (95% CI: 1.39, 1.50) in AFR to 3.51 (95% CI: 2.41, 5.10) in AI (Fig. 2B and Supplementary Data 5). Compared to individuals in the 40–60% PRS category, the top 90–100% PRS

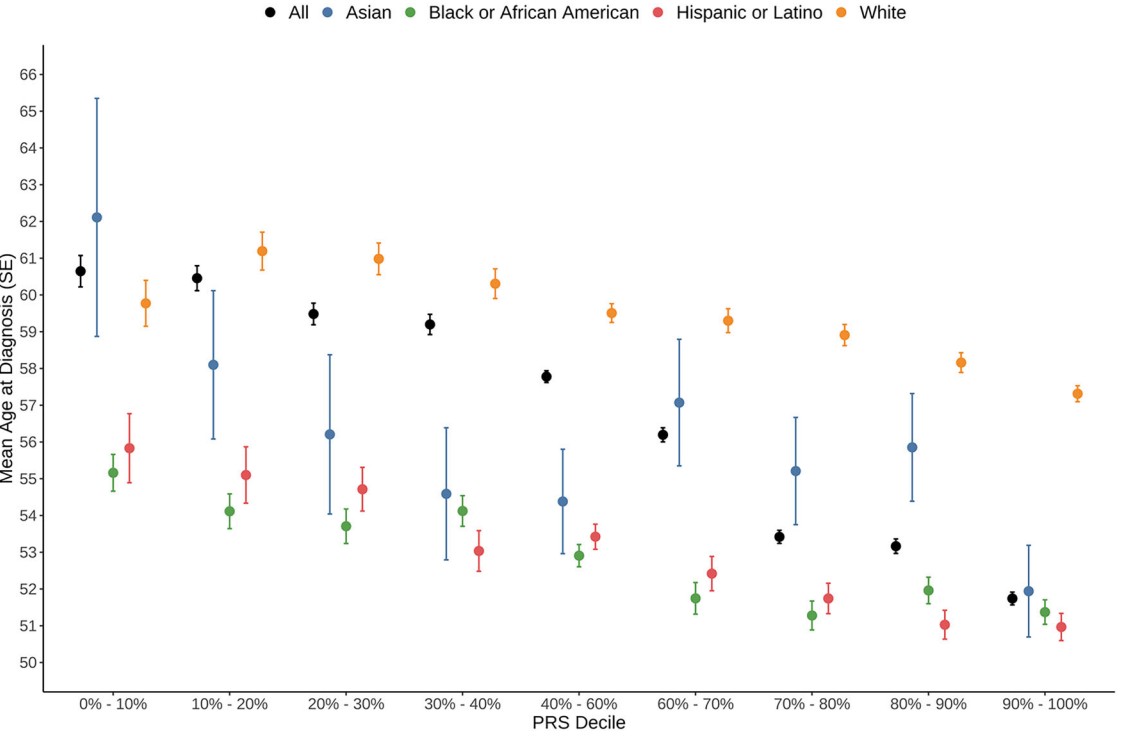

**Fig. 3 | Distribution of age at T2D diagnosis by PRS decile and population in All of Us.** Data are presented as mean age at diagnosis with standard error. Sample sizes are provided in Supplementary Data 6.

decile was associated with 2.37 (95% CI: 2.24, 2.51) increased odds of T2D across all populations combined, ranging from 1.75 (95% CI: 1.54, 2.00) in AFR to 6.96 (95% CI: 2.25, 21.56) in AI (Supplementary Fig. 2). Effects were generally similar in the 13 additional biobanks and cohorts within each population (Fig. 2).

As a secondary analysis, we evaluated the association between the T2D PRS and age at T2D diagnosis (Fig. 3 and Supplementary Data 6). Higher T2D PRS was significantly associated with younger age at diagnosis across populations. Individuals with T2D in the top 10% PRS decile were diagnosed, on average, 8.9 years earlier (95% CI: −9.83 to −7.89; $P = 1.09 \times 10^{-78}$) than those in the bottom 10% (Supplementary Data 6). This pattern was consistent across populations, with individuals in the top 10% PRS decile diagnosed on average 3.79 years earlier in AFR ($P = 1.48 \times 10^{-9}$), 10.17 years earlier in ASN ($P = 1.67 \times 10^{-3}$), 2.46 years earlier in EUR ($P = 3.33 \times 10^{-4}$), and 4.86 years earlier in HIS ($P = 8.75 \times 10^{-7}$) populations compared to those in the bottom 10% decile.

### Context-dependent effects of T2D PRS

We investigated whether the T2D PRS demonstrated context-dependent effects on T2D risk by conducting association analyses stratified by 18 demographic, medical, lifestyle, and behavioral characteristics, and evaluating effect heterogeneity using Cochran's $Q$-test and interaction terms in regression models (see the "Methods" section and Supplementary Data 7). Across PAGE and the additional biobanks and cohorts, we conducted stratified analyses in each of four populations (AFR, ASN, EUR, HIS) with sufficient sample sizes, as well as in all populations combined. We consistently found significant differences in T2D PRS effects by age groups, sex, hypertension status, and obesity status within and across populations (Fig. 4 and Supplementary Data 8–12).

Specifically, in meta-analyses across all populations, the T2D PRS was associated with significantly higher risk of T2D in younger individuals (age ≤50 years: OR = 1.91 (95% CI: 1.88, 1.94); age 50–60 years: OR = 1.85 (95% CI: 1.83, 1.88); age 60–70 years: OR = 1.81 (95% CI: 1.79,

1.83); age >70 years: OR = 1.68 (95% CI: 1.66, 1.71); $P_{het} = 4.20 \times 10^{-28}$). The T2D PRS also had a significantly larger effect in males compared to females (males: OR = 1.91 (95% CI: 1.90, 1.92); females: OR = 1.74 (95% CI: 1.72, 1.76); $P_{het} = 2.00 \times 10^{-38}$). The effect of T2D PRS was slightly higher in individuals without hypertension (without hypertension: OR = 1.87 (95% CI: 1.85, 1.90); with hypertension: OR = 1.83 (95% CI: 1.82, 1.85); $P_{het} = 8.30 \times 10^{-03}$). The effect of the T2D PRS was attenuated in those with BMI categorized as obese (non-overweight/obese: OR = 1.89 (95% CI: 1.86, 1.92); overweight: OR = 1.96 (95% CI: 1.94, 1.98); obese: OR = 1.76 (95% CI: 1.75, 1.78); $P_{het} = 9.19 \times 10^{-56}$). Although no significant PRS performance difference by family history (FH) of T2D was observed in meta-analyses across all populations, a stronger effect was observed among those with a FH of T2D in AFR populations (FH: OR = 1.56 (95% CI: 1.50, 1.63); without FH: OR = 1.49 (95% CI: 1.46, 1.52); $P_{het} = 0.04$) (Fig. 4, Supplementary Data 9). In analyses performed separately in PAGE versus the additional biobanks and studies, evidence of replication was observed for T2D PRS effects when stratifying by age groups, sex, hypertension and obesity status (Supplementary Fig. 3, Supplementary Data 10 and 11), as summarized in Supplementary Data 8. AUCs calculated separately in each stratum further supported these findings (Supplementary Data 10).

PRS had a stronger effect in never smokers and former smokers compared to current smokers ($P_{het} = 4.08 \times 10^{-18}$) and in physically active individuals compared to inactive individuals ($P_{het} = 6.14 \times 10^{-3}$) in meta-analyses across all populations, although these findings were not consistent when examining each population separately. Additionally, the PRS effect was stronger in statin and lipid-lowering medication users than non-users, though the differential association is complicated by higher statin use among diabetes cases and incomplete documentation of lipid medication use. Findings were inconsistent for other medication use, lipids, and current drinking status.

### Association between T2D PRS and diabetes-related traits

Next, to investigate the clinical implications of genetic risk of T2D, we examined associations between the T2D PRS and 20 diabetes-related

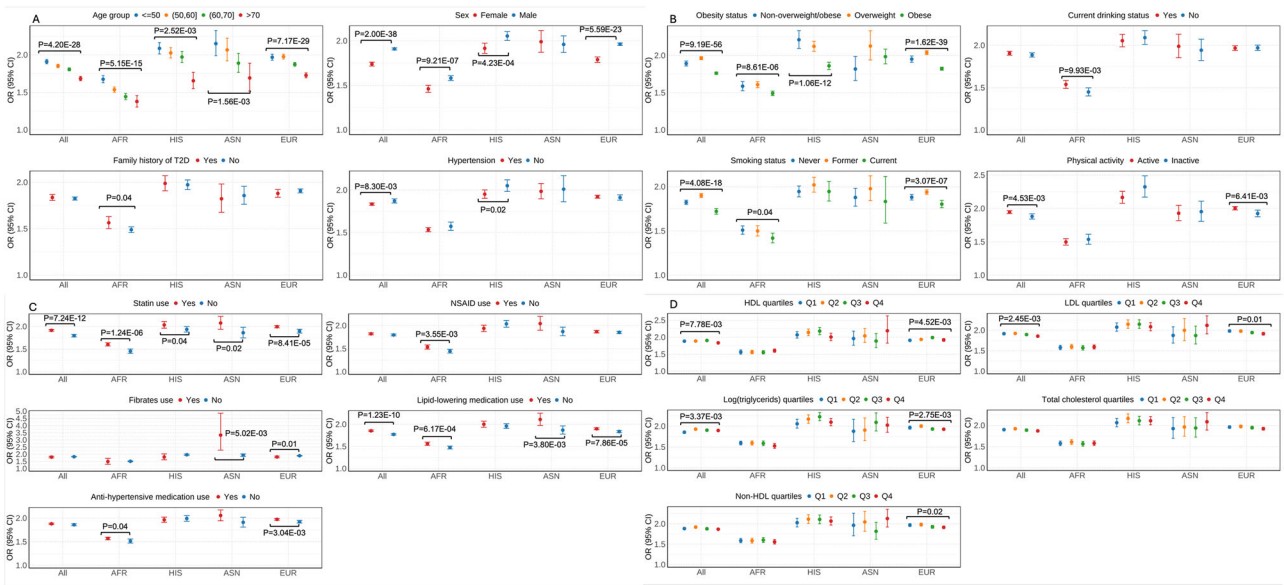

**Fig. 4 | Effect of the T2D PRS on T2D risk stratified by demographic, medical, and lifestyle and behavioral factors meta-analyzed across PAGE and the additional biobanks and cohorts.** Data are presented as odds ratio (OR) with 95% confidence interval (CI). *P*-values of heterogeneity from a two-sided Cochrane *Q*-test are indicated when differences were statistically significant (*P* < 0.05). **A** Demographic characteristics and medical history factors. **B** Behavioral and lifestyle factors. **C** Medication use. **D** Lipids. Sample sizes for **A**–**D** are provided in Supplementary Data 9.

traits across five primary phenotype categories: glycemic traits, cardiometabolic traits, vascular diseases, kidney diseases and functions, and inflammatory biomarkers, stratified by T2D status in AFR, ASN, EUR, HIS, and all populations combined (see the "Methods" section and Supplementary Data 13). Significant associations (applying a $P < 2.50 \times 10^{-3}$ [0.05/20 traits] Bonferroni adjusted threshold) were found across various glycemic and cardiometabolic traits, including HbA1c, fasting glucose, hypertension, systolic blood pressure (SBP), diastolic blood pressure (DBP), triglycerides, non-HDL cholesterol, and HDL cholesterol (Fig. 5, Supplementary Data 14), with consistent results observed in analyses performed separately in PAGE versus the additional biobanks and studies (Supplementary Data 15 and 16). AUC and $R^2$ estimates further supported these findings (Supplementary Data 15).

We first examined the associations between T2D PRS and glycemic traits among T2D controls, as glycemic measurements in T2D cases are influenced by diabetes medications. Each SD increase in the T2D PRS was significantly associated with 0.034 (95% CI: 0.031, 0.037) higher units of HbA1c levels and 0.030 mmol/L (95% CI: 0.026, 0.035) higher fasting glucose levels in individuals without T2D (Fig. 5A). These findings were consistent across populations and studies. Additionally, the PRS demonstrated a significant association with log transformed HOMA-IR ($P = 5.19 \times 10^{-4}$), although this was observed only in the EUR population.

Regarding cardiometabolic traits, each SD increase in PRS was associated with a 1.04-fold increased risk (95% CI: 1.03, 1.05) of hypertension in T2D controls across all populations, a 1.04-fold (95% CI: 1.02, 1.06) and 1.03-fold (95% CI: 1.02, 1.04) increased risk in EUR cases and controls, and a 1.10-fold (95% CI: 1.07, 1.13) increased risk in HIS controls (Fig. 5B). PRS was also positively associated with SBP ($P = 8.24 \times 10^{-25}$), DBP ($P = 2.72 \times 10^{-6}$), log-transformed triglycerides ($P = 1.13 \times 10^{-68}$), and non-HDL cholesterol ($P = 1.50 \times 10^{-5}$), and negatively associated with HDL cholesterol ($P = 1.50 \times 10^{-5}$) in T2D controls across all populations.

For kidney diseases, the PRS was associated with a 1.08-fold increased risk (95% CI: 1.03, 1.13) of end stage renal disease (ESRD) in T2D cases across all populations (Supplementary Fig. 4B). However, PRS was inversely associated with a 0.87-fold decreased risk (95% CI:

0.80, 0.95) of chronic kidney disease (CKD) in ASN controls. The PRS was not consistently associated with any vascular diseases or inflammatory markers.

**T2D PRS PheWAS**
To further investigate the prognostic value of the T2D PRS and clinical phenotypes, we conducted a comprehensive PheWAS meta-analysis of the T2D PRS on 1,815 unique phenotypes representing 17 disease categories tested across five biobanks. We identified significant associations between the T2D PRS and 732 traits (40.3% of all phecodes tested) across all populations combined at Bonferroni adjusted significance ($P < 2.75 \times 10^{-5}$), as well as 213 (12.0%), 43 (3.7%), 689 (38.0%), and 232 (13.7%) significant traits in AFR, ASN, EUR, and HIS populations, respectively (Fig. 6, Supplementary Fig. 5, and Supplementary Data 17). All significant traits identified in AFR, ASN, and HIS and were also significant in the EUR population, which had an additional 428 significant traits that were not significantly associated in any other population (Fig. 7). The PRS-PheWAS effect estimates (ORs) in EUR were reasonably correlated with AFR ($r = 0.68$), ASN ($r = 0.64$), and HIS ($r = 0.73$) populations, with consistent effect directions. Effects had slightly lower correlations between AFR and HIS ($r = 0.58$), ASN and AFR (0.58), and ASN and HIS (0.59), likely reflecting the larger sample size in the EUR population, which allowed more accurate estimates of effects and thus more overlap with other populations.

The 732 significant associations encompassed phenotypes related to the circulatory system ($n = 123$, 16.8%), endocrine and metabolic functions ($n = 90$, 12.3%), digestive system conditions ($n = 82$, 11.2%), sense organs ($n = 65$, 8.9%), genitourinary system conditions ($n = 58$, 7.9%), and 12 other disease categories ($n = 314$, 42.9%). In analyses conducted across all populations combined, T2D (OR = 2.38; *P*-value < $5.0 \times 10^{-324}$) demonstrated the most significant association, followed by T2D-related phenotypes from the endocrine/metabolic phenotype group, including diabetic retinopathy, insulin pump user, polyneuropathy in diabetes, and diabetes mellitus (Table 2 and Fig. 6). We observed consistently strong associations overall and across populations between the T2D PRS and type 1 diabetes (T1D) risk, with each SD increase in the T2D PRS associated with 1.63-fold ($P = 1.31 \times 10^{-173}$), 1.47-fold ($P = 1.31 \times 10^{-173}$), 1.83-fold ($P = 2.97 \times 10^{-18}$),

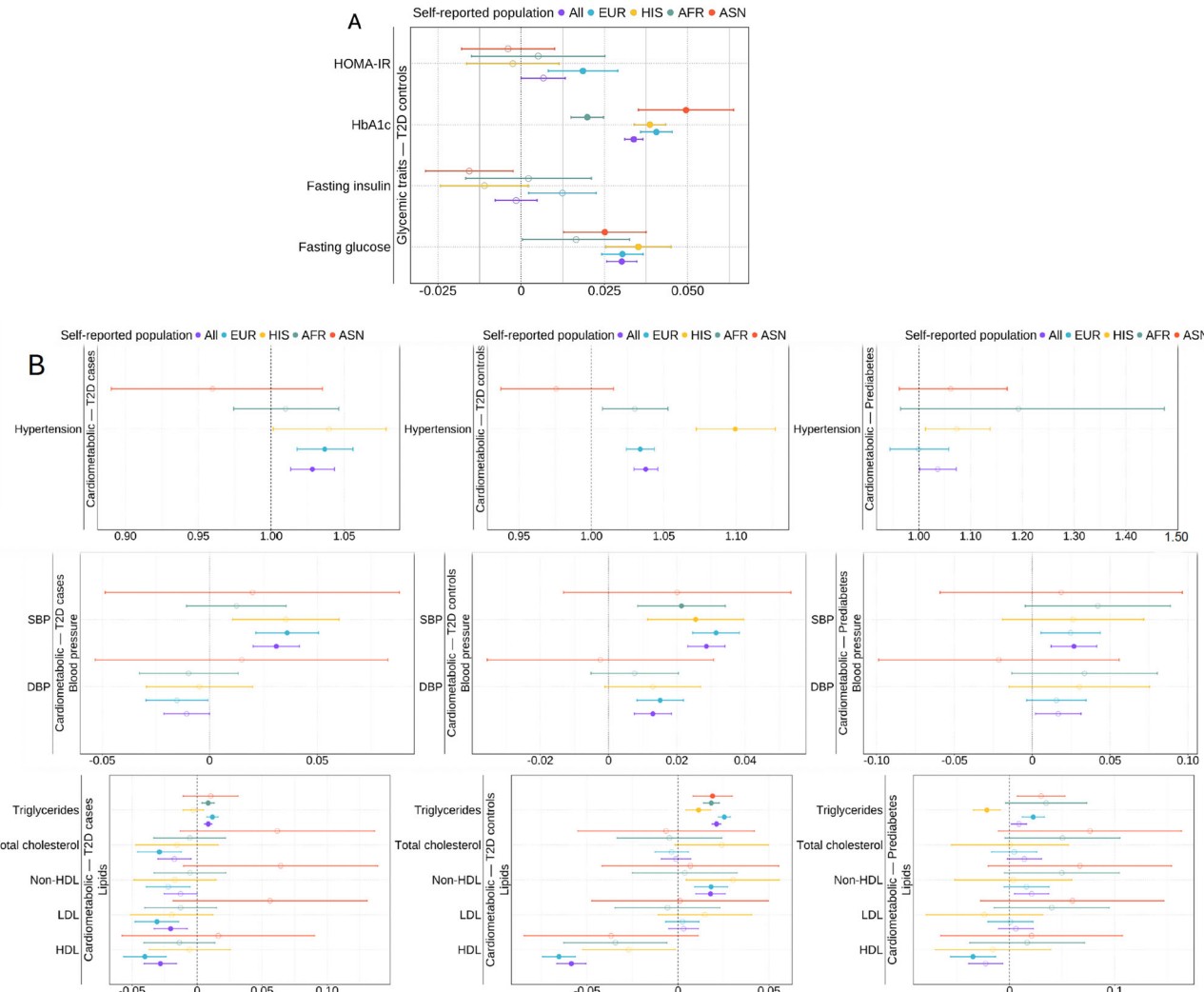

**Fig. 5 | Effect of T2D PRS on diabetes-related traits meta-analyzed across PAGE and the additional biobanks and cohorts.** The *X*-axis represents the beta estimates and 95% confidence intervals (CIs) of PRS for continuous traits and odds ratios (OR) and 95% CIs of PRS for binary traits. *P*-values shown on each plot are exact values derived from two-sided linear regression models for continuous traits and two-sided logistic regression models for binary traits. Solid circles indicate significant associations that passed the Bonferroni-adjusted *P*-value threshold of $P < 2.50 \times 10^{-3}$, while open circles indicate associations that were not significant. Panels show associations for the T2D PRS and **A** glycemic traits in T2D controls and **B** cardiometabolic traits in T2D cases (left), controls (middle), and individuals with prediabetes (right). The units of continuous traits are listed in Supplementary Data 13. Sample sizes are provided in Supplementary Data 14. Results for vascular disease, kidney function related traits, and inflammatory biomarkers are shown in Supplementary Fig. 4.

1.63-fold ($P < 5.0 \times 10^{-324}$), and 1.76-fold ($P = 4.41 \times 10^{-128}$), and increased risk of T1D in the overall, AFR, ASN, EUR, and HIS populations, respectively.

The single top phenotype associated with the T2D PRS from other phenotype groups, all of which were positively associated, were essential hypertension, CKD, senile cataract, chronic ulcer of leg or foot, anemia in chronic kidney disease, acute osteomyelitis, other peripheral nerve disorders, open wound of foot, respiratory failure, edema, gastroparesis, tobacco use disorder, diabetes or abnormal glucose tolerance complicating pregnancy disorder, benign neoplasm of colon, dermatophytosis, and cardiac congenital anomalies.

## Discussion

In this large-scale T2D investigation of more than 240,000 T2D cases and 630,000 controls from PAGE and 13 independent biobanks and studies, we performed the first comprehensive and systematic cross-population characterization of a multi-ancestry genome-wide T2D

PRS. This was achieved using a harmonized framework to evaluate context-dependent T2D PRS effects across different demographic and clinical characteristics, T2D PRS associations with diabetes-related traits, and a PheWAS of the T2D PRS. Our findings demonstrate that the PRS holds predictive value for T2D risk across diverse populations and was associated with younger age at T2D diagnosis. The T2D PRS demonstrated context-dependent associations with T2D risk, with PRS performance varying by certain demographic and clinical characteristics, such as age, sex, hypertension, and obesity status. Additionally, we found that the genetic risk of T2D, as measured by the T2D PRS, was associated with other glycemic and cardiometabolic traits in controls, suggesting that PRS could provide insights into T2D etiology and serve as a valuable tool for predicting dysglycemia and metabolic risk, thereby facilitating early interventions and improving clinical risk assessment. Notably, associations between the T2D PRS and diabetes-related traits and complications were often consistent across populations, despite variable between-population PRS performance in estimating T2D risk, suggesting that the PRS may have clinical and

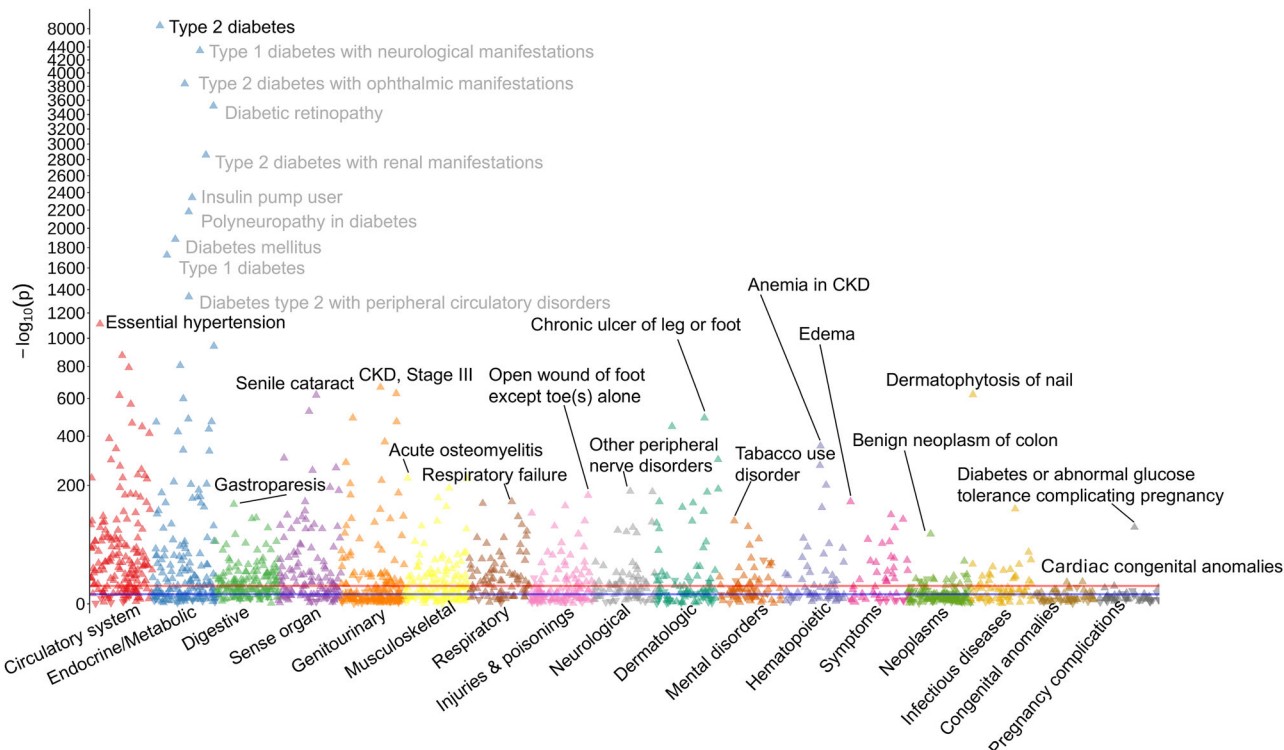

**Fig. 6 | T2D PRS PheWAS results meta-analyzed across all five biobanks and populations.** The *X*-axis represents phecodes color-coded by their corresponding phenotype category and is ordered by the category with the most to the category with the least significant hits. The *Y*-axis represents the −log₁₀(*p*-value) derived from two-sided logistic regression models. The red horizontal line represents Bonferroni-adjusted *p*-value threshold $P < 2.75 \times 10^{-5}$ (732 phecodes meet this threshold), and the blue horizontal line represents an unadjusted *p*-value threshold of $P < 0.05$ (1120 phecodes meet this threshold). Upward triangles indicate positive associations, while downward triangles indicate negative associations. The top ten most significant associations from the endocrine/metabolic category are annotated, with the single most significant association shown in black and others shown in gray, while the single most significant association from each other category is annotated.

prognostic utility beyond T2D risk prediction. Our PheWAS found associations between the T2D PRS and 40.3% of tested phenotypes, including phenotypes across all disease categories tested, underscoring the pleiotropic effects of T2D genetic risk and suggesting a broad biological impact of T2D-associated variants across organ systems. By providing a comprehensive resource that characterizes T2D PRS across diverse populations, our findings move beyond European-centric discovery and evaluate whether PRS utility—both predictive and prognostic—depends on individual- and population-level contexts. Future applications of T2D PRS in precision medicine should consider not only ancestry-specific calibration but also how environmental, metabolic, and clinical factors modulate the phenotypic expression of genetic risk.

We found that a multi-ancestry PRS was strongly associated with increased risk of T2D in all populations examined, yielding an overall OR of 1.75 (95% CI: 1.71, 1.78) and AUC of 0.641 (95% CI: 0.636, 0.645). The PRS performance across studies was generally comparable, with small variations likely due to study-specific differences, such as study design and phenotype definition. However, we observed larger variations in the performance of T2D PRS across populations, with the highest predictive performance in the AI population (although sample sizes were small), followed by EUR, ASN, NH, HIS, and AFR populations. These findings are consistent with what has been previously reported[8,16] and could be attributed to various population-specific factors that are likely not strongly represented in the discovery GWAS used to construct the T2D PRS[20]. This notably underscores the importance of expanding GWAS and fine-mapping efforts, particularly in AFR populations, to improve the accuracy and transferability of T2D PRS.

Despite the lower predictive ability of the T2D PRS in AFR and HIS populations, we observed the strongest evidence of the PRS being associated with a younger age at T2D diagnosis in these two populations, demonstrating potential clinical utility. AFR and HIS populations tended to develop diabetes earlier than ASN and EUR populations; while these differences could be partially attributed to the younger age at enrollment in these populations (mean age at enrollment: AFR = 54.7 [SD: 10.2], HIS = 53.3 [SD: 11.4], ASN = 54.5 [SD: 12.5], EUR = 60.5 [SD: 12.5]), similar differences in age at T2D diagnosis have been previously reported[21]. As such, these differences may reflect complex interactions between genetic and non-genetic factors that may warrant further studies to understand the causes and mechanisms of earlier T2D onset in these groups.

Our results show that the prediction accuracy of PRS is context-dependent and can vary even within populations. We consistently observed that the T2D PRS was particularly informative for predicting T2D susceptibility in younger individuals, those without hypertension, males, and individuals who were not obese or overweight, across and within populations. The stronger predictive ability in younger individuals is consistent with published literature, such as the findings from the Framingham Offspring Study[22] and the EPIC InterAct Study[9], both of which observed a higher relative effect of T2D PRS among individuals under 50 years old compared to those over 50. However, these studies have been limited to T2D PRS developed and evaluated in individuals predominantly of European descent. Similar trends have also been observed in other traits, including prostate cancer[23–25], colorectal cancer[26,27], estrogen receptor-positive breast cancer[28], and coronary artery disease[29]. The stronger genetic influence on disease onset at a younger age is expected, as older individuals have had more

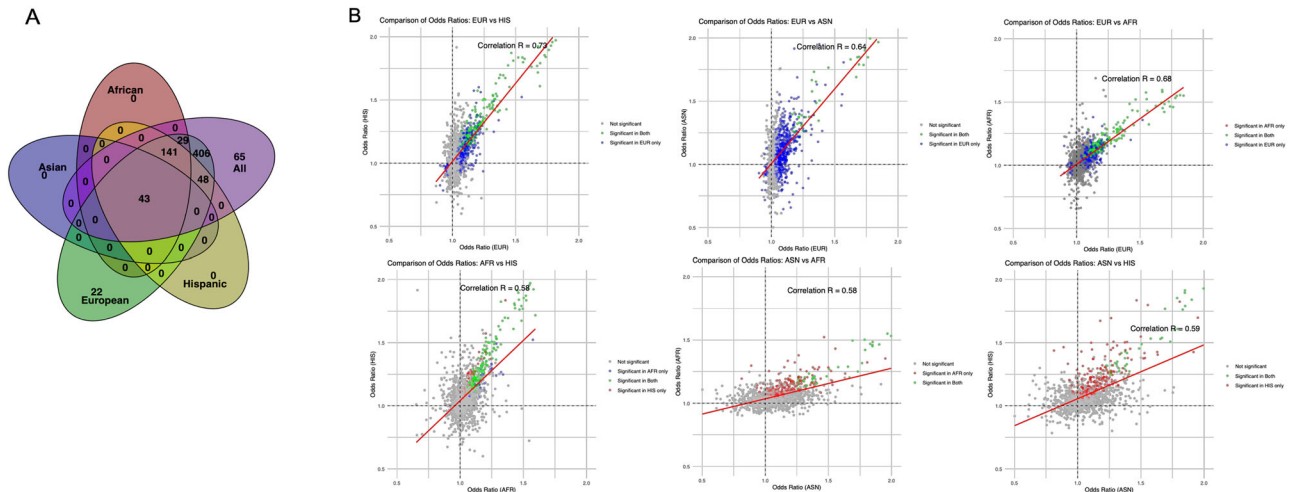

**Fig. 7 | Comparison of significant associations and effect sizes of T2D PRS PheWAS results by population. A** Venn diagram of significant PheWAS results across populations. **B** Comparison of odds ratios from T2D PRS PheWAS results across populations.

time to accumulate environmental risk factors that lead to metabolic impairments, thereby impacting disease risk. The larger effect of T2D PRS in leaner adults and those without hypertension could similarly be attributed to a lower accumulation of environmental risk factors in these individuals, suggesting a higher contribution of genetic risk factors to T2D development, a pattern also shown in several previous studies[8,9,30].

Prior research on the performance of T2D PRS stratified by sex is limited[9,31], and evidence of sex-based differences in T2D heritability estimates is inconsistent[32–35]. However, sex-based differences in PRS prediction have been widely reported across a range of complex traits[36–38]. In this study, we observed stronger predictive performance of the T2D PRS in males. This difference could be attributed to several factors. First, certain genetic variants exhibit sex-dimorphic effects, with some loci demonstrating larger or more penetrant effects in males—for example, variants at *IRS1* show male-specific effects on fasting insulin and the 11p15.5 region, which harbors genes involved in insulin production such as *KCNQ1*, has been implicated in male-specific T2D risk, potentially influenced by maternal inheritance patterns[39–42]. Second, sex-specific biological mechanisms, including differences in glucose and lipid metabolism, affect fat distribution and insulin sensitivity, which can modulate the penetrance of these genetic variants[41,42]. Additionally, diagnosis bias may lead to T2D being more frequently diagnosed in males[41]. Future sex-specific GWAS and gene-by-sex interaction studies are needed to further investigate sex-specific genetic factors in T2D.

Collectively, these interactions between the T2D PRS and non-genetic factors demonstrate the complex nature of T2D risk and the importance of interpreting the genetic risk of T2D in the context of non-genetic risk factors. Our findings suggest that across diverse populations, genetic effects can vary across contexts and that PRS may exhibit context-specific performance. This indicates that risk prediction models will need to include not only PRS but demographic, medical, lifestyle and behavioral characteristics, and their interactions[36]. Moreover, any potential application of PRS in clinical care will need to consider such contextual factors.

We found that the T2D PRS was associated with various diabetes-related traits and disease conditions across diverse populations, providing compelling evidence that the T2D PRS could provide insights into T2D prognosis. We observed consistent associations between T2D PRS and glycemic parameters in T2D controls, with significant positive associations for fasting glucose and HbA1c, as well as associations with cardiometabolic traits, including hypertension, SBP, triglycerides, and HDL cholesterol, in both T2D cases and controls. These findings

suggest that T2D PRS may serve as a valuable tool for early identification of individuals at risk for developing dysglycemia and cardio-metabolic complications, even among those who are currently asymptomatic or undiagnosed. Our findings also have important implications for T2D etiology and pathophysiology. For instance, the positive association between T2D PRS and HOMA-IR in EUR controls provides genetic evidence that complements previous epidemiological studies on the critical role of insulin resistance in T2D development[43–45]. This association suggests that genetic factors contribute to insulin resistance pathways even before clinical diabetes onset, offering mechanistic insights into how genetic predisposition translates into disease risk. The inverse association we observed between T2D PRS and CKD in ASN controls warrants further investigation, as it could reflect selection bias where individuals with higher genetic risk for T2D who remain diabetes-free may represent a healthier subset, possibly benefiting from protective environmental or lifestyle factors that also lower their CKD risk.

Findings from our PheWAS across 1815 diseases and conditions and five biobanks further demonstrated significant associations between T2D PRS and various clinical comorbidities, including those related to the circulatory system, endocrine and metabolic disorders, lipid abnormalities, and kidney diseases. Notably, these associations were consistently observed across populations, with fairly correlated effect estimates. Despite the variability in PRS performance for T2D risk estimation across populations, our findings suggest that genetic mechanisms underlying T2D and diabetes-related traits are shared across populations. As such, the PRS may serve as a valuable tool for stratifying the risk of complications in individuals diagnosed with T2D across diverse populations. The positive association between T2D PRS and micro- and macrovascular complications, such as retinopathy, neuropathy, CKD, cardiovascular diseases, and peripheral artery disease, has also been reported by prior studies[7,46,47]. The significant positive associations between the T2D PRS and T1D risk from PheWAS may indicate the potential misclassification of T2D and T1D in clinician-recorded diagnoses, given the overlapping clinical features of T2D and T1D[48], or it could reflect genetic consistencies between the two diseases. This study complements prior genetics research on T2D by leveraging large sample sizes to identify significant associations between T2D PRS and various traits and conditions not previously reported. Our findings imply that the T2D PRS holds promise as a powerful tool for precision medicine, risk stratification, and disease management in the context of T2D, providing an opportunity to identify individuals at elevated risk of developing T2D and its complications. Further, the PheWAS results reported here serve as a

**Table 2 | Significant meta-analyzed phenome-wide associations (meta-PheWAS) between T2D PRS and clinical diseases across all populations**

| Phecode | Phenotype | Disease category | OR | P-value |
|---|---|---|---|---|
| 250.2 | Type 2 diabetes | Endocrine/Metabolic | 2.38 | <5E–324 |
| 250.7 | Diabetic retinopathy | Endocrine/Metabolic | 2.52 | <5E–324 |
| 250.3 | Insulin pump user | Endocrine/Metabolic | 2.19 | <5E–324 |
| 250.6 | Polyneuropathy in diabetes | Endocrine/Metabolic | 2.23 | <5E–324 |
| 250 | Diabetes mellitus | Endocrine/Metabolic | 1.80 | <5E–324 |
| 250.1 | Type 1 diabetes | Endocrine/Metabolic | 2.05 | <5E–324 |
| 272.1 | Hyperlipidemia | Endocrine/Metabolic | 1.27 | <5E–324 |
| 249 | Secondary diabetes mellitus | Endocrine/Metabolic | 2.05 | <5E–324 |
| 250.4 | Abnormal glucose | Endocrine/Metabolic | 1.36 | <5E–324 |
| 276.13 | Hyperpotassemia | Endocrine/Metabolic | 1.36 | <5E–324 |
| 401.1 | Essential hypertension | Circulatory system | 1.34 | <5E–324 |
| 536.3 | Gastroparesis | Digestive | 1.46 | 6.66E–142 |
| 366.2 | Senile cataract | Sense organs | 1.27 | <5E–324 |
| 585.3 | Chronic renal failure | Genitourinary | 1.34 | <5E–324 |
| 710.11 | Acute osteomyelitis | Musculoskeletal | 1.65 | 1.55E–225 |
| 509.1 | Respiratory failure | Respiratory | 1.19 | 1.44E–149 |
| 871.3 | Open wound of foot except toe(s) alone | Injuries & poisonings | 1.49 | 2.97E–168 |
| 351 | Other peripheral nerve disorders | Neurological | 1.16 | 5.25E–181 |
| 707.2 | Chronic ulcer of leg or foot | Dermatologic | 1.54 | <5E–324 |
| 318 | Tobacco use disorder | Mental disorders | 1.09 | 1.07E–98 |
| 285.21 | Anemia in chronic kidney disease | Hematopoietic | 1.50 | <5E–324 |
| 782.3 | Edema | Symptoms | 1.15 | 2.91E–149 |
| 208 | Benign neoplasm of colon | Neoplasms | 1.08 | 7.55E–70 |
| 110.1 | Dermatophytosis | Infectious diseases | 1.10 | 1.38E–38 |
| 747.1 | Cardiac congenital anomalies | Congenital anomalies | 1.06 | 1.45E–07 |
| 649.1 | Diabetes or abnormal glucose tolerance complicating pregnancy | Pregnancy Complications | 1.46 | 6.32E–84 |

If a Phecode with the same one-decimal value (e.g., 250.2) is listed, more specific phecodes (e.g., 250.24) are excluded; otherwise, the specific phecode is shown. The top ten associations in the endocrine/metabolic disease category are shown, while the top one association is shown for other phenotype categories. All associations shown passed Bonferroni-adjusted p-value threshold $P < 2.75 \times 10^{-5}$. Full meta-PheWAS results are provided in Supplementary Data 17.

comprehensive catalog of phenotypic associations with T2D PRS, contributing to our understanding of T2D-related comorbidity patterns. These findings could offer a valuable resource for hypothesis generation, disease subtype characterization, and the identification of shared biological pathways and molecular mechanisms.

The results of our study should be interpreted considering several limitations. We used self-identified race and ethnicity (SIRE) population descriptors to determine whether the T2D PRS had discrepancies in performance in the contexts of SIRE. Although other sociodemographic and environmental factors may be more informative than SIRE, because our investigation included a large number of studies, many of which include legacy data collected up to 37 years ago, this study was limited by the variables available across all included studies. Further, SIRE information was collected differently between studies (Supplementary Data 2), which could influence PRS performance between populations. In addition, this study included both incident and prevalent T2D. Future research should replicate these findings by investigating the association between PRS and risk of incident T2D and the other outcomes reported here. Further, most phenotypes and clinical measurements, such as glucose levels and BMI, were collected at baseline, when these measurements are typically available for most individuals. This limits our ability to capture changes over time and assess the dynamic nature of these associations.

In conclusion, our large-scale, systematic and cross-population investigation highlights the predictive value of the T2D PRS across diverse populations, emphasizing its potential as a promising tool for identifying individuals at elevated risk of developing T2D and its associated comorbidities and complications, shedding light on potential avenues for risk prediction and personalized management strategies. Our findings also demonstrate that the T2D PRS has context-dependent effects, indicating the need to account for these characteristics to interpret PRS and improve PRS prediction accuracy. Further research is warranted to evaluate and optimize the clinical utility of T2D PRS across diverse populations.

## Methods
### Participants
The PAGE Study was developed to conduct genetic epidemiological research in ancestrally diverse populations within the United States[49]. PAGE includes individuals from the following ongoing population-based biobanks and cohorts: Atherosclerosis Risk in Communities (ARIC) Study[50], the Icahn School of Medicine at Mount Sinai BioMe biobank in New York City (BioMe), Coronary Artery Risk Development in Young Adults Study (CARDIA)[51], Hispanic Community Health Study/ Study of Latinos (HCHS/SOL)[52,53], Multiethnic Cohort Study (MEC)[54], and Women's Health Initiative (WHI)[55]. A detailed description of PAGE has been described[49] with additional detail summarized in the Supplemental Information and Supplementary Data 1.

Analyses were also conducted in 13 additional biobanks and cohorts that were not included in the discovery GWAS summary statistics of the best-performing T2D PRS. Studies included All of Us (AoU)[56], BioMe (participants were non-overlapping with those from BioMe included in PAGE), BioVU Biobank (BioVU)[57], Cameron County Hispanic Study (CCHC)[58], Cebu Longitudinal Health and Nutrition

Survey (CLHNS)[59], China Health and Nutrition Survey (CHNS)[60], Colorado Center for Personalized Medicine Biobank (CCPM)[61], Greenlandic Health Surveys (Greenlandic)[62–64], Multi-Ethnic Study of Atherosclerosis Study (MESA)[65], Michigan Genomics Initiative (MGI)[66], Million Veteran Program (MVP)[67–69], Qatar Biobank (Qatar)[70], and REasons for Geographic and Racial Differences in Stroke Study (REGARDS)[71] (Supplementary Data 1). Analyses performed in these additional studies were dependent on the availability of variables and data needed to derive variables.

Our research complies with all relevant ethical regulations regarding the use of human study participants and was conducted in accordance to the criteria set by the Declaration of Helsinki. All participants provided informed consent, and each study was approved by the appropriate Institutional Review Board at their respective institutions. Additional information about each participating study can be found in the Supplementary Information.

## Use of population descriptors
Self-identified race and ethnicity (SIRE) population descriptors were used across studies, with the goal of studying how PRS performance may vary across the contexts of SIRE. Although other sociodemographic and environmental factors are expected to be more informative than SIRE, this investigation is limited by the variables that are currently available across all included studies. It is important to note that SIRE does not serve as a proxy for genetic ancestry, holds no biological basis, and does not imply a biological explanation for health disparities[72].

We standardized population descriptors across all studies in the analysis by using consistent acronyms. Specifically, AFR refers to African, African American, or Black participants; AI refers to American Indian, Alaskan Native, or Native American; ASN includes Asian American, East or Southeast Asian (e.g., from China, Japan, Korea, Indonesia, the Philippines), as well as South Asian/Indian (e.g., from India, Pakistan); EUR refers to non-Hispanic White American or European individuals; Greenlandic refers to individuals of Greenlandic or mixed Greenlandic-Danish heritage; HIS represents Hispanic/Latino American participants; MENAQ encompasses Qatari and Arab American individuals; and NH refers to Native Hawaiian or Other Pacific Islander. The Greenlandic population was considered separate from others, as this population represents substantial Inuit ancestry[63]. Detailed definitions of population descriptors used in each study are provided in Supplementary Data 2.

## T2D and prediabetes definition
T2D and prediabetes definitions were created based on the American Diabetes Association's 2021 Standards of Medical Care in Diabetes[73]. In PAGE, participants were classified as T2D cases if they met at least one of the following criteria: adults ≥25 years old with (1) a T2D diagnosis by a physician/medical professional or use of medication for treatment of diabetes, (2) a fasting (≥8 h) plasma glucose ≥ 126 mg/dl, (3) a random glucose ≥ 200 mg/dl, or (4) an HbA1c ≥ 48 mmol/mol (6.5%). Cases were restricted to those with either an age at diagnosis or most recent age at glucose or HbA1c draw ≥25 years to avoid misclassifying T1D cases as T2D cases. Individuals with prediabetes were defined as adults ≥18 years old who did not meet the definition of T2D and with (1) a fasting plasma glucose between 100–125 mg/dl, (2) an HbA1c between 39 and 46 mmol/mol (5.7–6.4%), or (3) a 2-h oral glucose tolerance test between 140 and 199 mg/dl. T2D controls were defined as adults ≥ 40 years old who did not meet the definitions of T2D or prediabetes. Some studies, however, used their own in-house definitions and did not identify individuals with prediabetes. Definitions used in the additional biobanks and cohorts were similar to those used in PAGE and are summarized in the Supplemental Information.

## Genotyping, imputation, and quality control
This study included 36,724 PAGE participants who met the T2D and prediabetes definitions and underwent genotyping using the Multi-Ethnic Genotyping Array (MEGA) panel, specifically designed to capture genetic heterogeneity across diverse populations, with the remaining 46,220 PAGE participants genotyped using a variety of other arrays (Supplementary Data 1)[74]. Data were imputed using the NHLBI Trans-Omics for Precision Medicine (TOPMed) Imputation Server (https://imputation.biodatacatalyst.nhlbi.nih.gov/#!) using TOPMed-r2 as reference panel, Eagle v2.4 for phasing, and Minimac4 for imputation[69,75,76]. Variants were excluded if the imputation info score was <0.4 or if the effective sample size was <30. The effective sample size was calculated as $2 \times MAF \times (1-MAF) \times N \times INFO$, where MAF represents the minor allele frequency, $N$ is the sample size, and INFO is the imputation quality score. Principal component analysis was performed separately for each study. Principal components (PCs) were generated using EIGENSOFT[77–79]. The top 10 PCs were used to account for population structure in subsequent analyses.

## Construction of T2D PRS
Participants included in this investigation were not part of the discovery GWAS used to develop the T2D PRS. PRS for each participant were constructed as the weighted sum of effect alleles, weighted by variant-specific log ORs using "--score" option with "cols = + scoresums" modifier in PLINK 2.0[80]. All T2D PRS were standardized by subtracting the mean PRS and dividing by the standard deviation of PRS calculated in controls within each population. Therefore, 1-unit increase in PRS corresponds to a 1-standard deviation increase within each population.

The R package "rtracklayer"[81] was used to align and convert between the GRCh37 (hg19) and the GRCh38 (hg38) genome reference builds across genetic datasets. In the function "GRangesFromDataFrame", we included a strand field (strand.field = "strand"), which generated the output built with a strand column. If the forward (plus) strand became the reverse (minus) strand in the new build[82], the strand column would change from "+" to "−". The alleles were on the new build, then changed to the complement of the original input build.

PRS-CSx estimates population-specific and multi-ancestry PRS by jointly modeling discovery GWAS summary statistics and precalculated linkage disequilibrium (LD) reference panels based on the 1000 Genomes Project (1KG) Phase 3 participants from AFR, Admixed American (AMR), EAS, EUR, and SAS population samples[12]. The first three PRS methods used the PRS-CSx "auto" option, which automatically learns the global shrinkage parameter $\phi$ from the discovery summary statistics. The META = T option in PRS-CSx was used to generate multi-ancestry PRS using an inverse-variance-weighted meta-analysis of the population-specific posterior effect size estimates (*PRS Method 1* and *PRS Method 2*). The META = F option was used to generate five population-specific PRS, which were subsequently calculated in PAGE, standardized to a mean of 0 and SD of 1 by population and study, and used to calculate a linear combination PRS (*PRS Method 3*). In *PRS Method 3*, for each population and study in PAGE, the dataset underwent a three-fold cross-validation process, where the data was evenly divided into three equally sized subsets. In each iteration, one dataset was selected as the testing dataset, while the remaining two were used as the validation dataset. This process was repeated three times to evaluate all possible combinations of validation and testing datasets, so that each individual participated in both the validation and testing sets.

The following logistic regression model of the standardized population-specific PRS was fit in the validation dataset for each

population:

$$\text{logit(p)} \sim \beta_0 + \beta_{HIS} \times \text{PRS}_{HIS} + \beta_{AFR} \times \text{PRS}_{AFR} + \beta_{EAS} \times \text{PRS}_{EAS} \\ + \beta_{EUR} \times \text{PRS}_{EUR} + \beta_{SAS} \times \text{PRS}_{SAS}, \tag{1}$$

where logit($p$) is the log odds of having T2D and $\beta_0$ is the intercept, with each subsequent $\beta$ representing the regression coefficient for each population-specific PRS. If any of the regression coefficients were <0, the corresponding population-specific PRS and regression coefficient were removed from the regression model, and the model was refitted until all remaining coefficients were >0. The regression coefficients learned in the validation dataset were then used as weights ($w_k$) in the testing datasets to calculate the final linear combination PRS for each individual:

$$\text{Final linear combination PRS} = w_{HIS} \times \text{PRS}_{HIS} + w_{AFR} \times \text{PRS}_{AFR} \\ + w_{EAS} \times \text{PRS}_{EAS} + w_{EUR} \times \text{PRS}_{EUR} \tag{2} \\ + w_{SAS} \times \text{PRS}_{SAS}$$

We assessed the predictive accuracy of the four T2D PRS within each population in PAGE using AUCs. AUC values were calculated using the R package "pROC"[83] with logistic regression of T2D risk after adding PRS to a base model of age, sex, BMI, the first ten PCs of ancestry, and study. The best-performing PRS was calculated in the additional biobanks and cohorts (that were independent of the discovery GWAS used to develop the best-performing PRS) and was used for all subsequent analyses.

### Association between T2D PRS and age at T2D diagnosis
We evaluated the association between the T2D PRS and age at T2D diagnosis in a case-only analysis in All of Us. This analysis was limited to All of Us due to its large sample size, inclusion of multiple populations, and the ability to estimate age at diagnosis. Age at diagnosis was determined by the earliest occurrence start date of T2D mellitus (SNOMED concept ID: 201826), T2D mellitus without complication (SNOMED concept ID: 4193704), or disorder due to T2D mellitus (SNOMED concept ID: 443732). We tested the association between each PRS category (predictor) and age at diagnosis (outcome) in linear regression models, using the 0–10% PRS decile as the reference. Analyses were performed separately in each population and across all populations.

### Stratification analyses
We investigated the context-dependent effects of the best-performing T2D PRS in analyses stratified by demographic characteristics and medical history (age group, self-reported sex at birth, family history (FH) of T2D, and blood pressure categories), behavioral and lifestyle factors (obesity status, current drinking status, smoking status, and physical activity level), medication use (statin, NSAID, fibrates, lipid-lowering medication, and anti-hypertensive medication), and lipid traits (HDL, LDL, triglycerides, total cholesterol (TC), and non-HDL quartiles) (definitions provided in Supplementary Data 7). Analyses were conducted in all populations combined and separately in each population using logistic regression models that predict T2D status, adjusting for age, sex (except in sex-stratified analyses), BMI, the first ten PCs, and study. AUCs were also calculated within each stratum using similar models, as well as models excluding PRS and models of PRS alone. The Cochrane $Q$-test was used to determine whether effect heterogeneity between subgroups was sufficiently large enough to be attributed to factors beyond sampling error. The $P$-value from the Cochrane $Q$-test is referred to as $P_{het}$. A $P < 0.05$ threshold was used to determine statistical significance for stratification analyses. We further evaluated evidence of context-dependent effects of PRS within PAGE by including an interaction term between the PRS and each context variable, also adjusting for the main effects of each. A likelihood ratio test was used to compare models with and without the interaction term.

### T2D PRS associations with diabetes-related traits
We next examined the ability of the best performing T2D PRS to predict 20 traits and complications, including glycemic traits (fasting glucose, fasting insulin, HOMA-IR, and HbA1c), cardiometabolic traits (HDL, LDL, triglycerides, TC, non-HDL, systolic and diastolic blood pressure, and hypertension), vascular diseases (heart attack and stroke), kidney diseases and functions (end stage renal disease, chronic kidney disease, urine albumin-creatine ratio, and eGFR), and inflammatory biomarkers (CRP and IL6) (Supplementary Data 13). Analyses were performed in all populations combined and separately in each population using regression models adjusting for age, sex, BMI, the top ten PCs, and study. Analyses of glycemic traits were restricted to T2D controls to mitigate confounding effects from diabetes medications in T2D cases. For the rest of the traits, analyses were conducted separately in T2D cases, T2D controls, and prediabetes individuals. Model performance was evaluated using AUCs and odds ratios (ORs) for dichotomous traits and the proportion of variance explained ($R^2$) and beta coefficients for continuous traits. Statistical significance was defined as $P < 2.50 \times 10^{-3}$ (0.05/20 traits).

### PheWAS
To further understand the clinical implications of T2D PRS on the risk of other diseases and conditions, we conducted phenome-wide association studies (PheWASs) in five independent biobanks: BioMe, BioVU, CCPM, MVP, and MGI. Utilizing the PheWAS R package[84], we ran logistic regression models to assess PRS-phenotype associations, adjusting for age, sex, BMI, and the first ten PCs. For the mapping of the International Classification of Diseases (ICD) 9 and ICD-10 codes to phecodes, we utilized phecodes version 1.2 for ICD-9-CM and the 2018 beta version of ICD-10-CM. Cases were considered valid if they had a minimum phecode count of two. PheWAS analyses were performed separately in each population and study, and a pooled-sample PheWAS was conducted with adjustments for populations in each study. Only phecodes with at least 10 cases were considered. A total of 1815 unique phenotypes across 17 disease categories were tested across the five biobanks, with 1777, 1171, 1813, and 1695 phenotypes available in AFR, ASN, EUR, and HIS populations, respectively. Significance was defined by correcting for the total number of phenotypes tested in each population using the Bonferroni-adjusted $P$-value thresholds: $P < 2.75 \times 10^{-5}$ for all populations combined, $P < 2.81 \times 10^{-5}$ for AFR, $P < 4.27 \times 10^{-5}$ for ASN, $P < 2.76 \times 10^{-5}$ for EUR, and $P < 2.95 \times 10^{-5}$ for HIS populations. Any $P$-values smaller than the smallest non-zero value that R can represent are denoted as $<5.0 \times 10^{-324}$.

### Meta-analysis
Results from PAGE and the 13 additional biobanks and cohorts were first meta-analyzed using inverse variance weighted fixed-effect models within populations to obtain population-specific results for AFR, ASN, EUR, and HIS populations. The results from these four populations, along with the results from all other populations (AI, Greenlandic, MENAQ, and NH), were included in the larger meta-analysis conducted among all populations combined. PheWAS results were similarly meta-analyzed across the five biobanks for each of the four populations and across all populations combined. The degree of heterogeneity was evaluated using the $\tau^2$ (between-study variance) and $I^2$ (total proportion of variance owing to heterogeneity) statistics. Meta-analyses were conducted only for analyses with at least 100 cases and 100 controls. If only one study was available for an analysis, making it unfeasible to meta-analyze with other studies, we presented results from that single study.

All statistical analyses were performed using R (version 4.1.2).

## Reporting summary

Further information on research design is available in the Nature Portfolio Reporting Summary linked to this article.

## Data availability

The Ge et al. T2D PRS [https://doi.org/10.1186/s13073-022-01074-2] and the T2D PRS of 582 variants [https://doi.org/10.1038/s41588-020-0637-y; https://doi.org/10.1016/j.xhgg.2021.100029] are available on the PGS Catalog under accession numbers: PGS002308 and PGS000804). The newly generated PRS in this study are published on the PGS Catalog under accession number: PGP000742). PheWAS results generated in this study are published on Zenodo [https://doi.org/10.5281/zenodo.15998801]. GWAS summary statistics from the Mahajan et al. 2022 study [https://doi.org/10.1038/s41588-022-01058-3] are available on the DIA-GRAM consortium website [https://diagram-consortium.org/downloads.html]. GWAS summary statistics from the Vujkovic et al. 2020 study [https://doi.org/10.1038/s41588-020-0637-y] are available on dbGaP under accession numbers: phs001672.v12.p1. LD matrices used for PRS-CSx are available at https://github.com/getian107/PRScsx.

## Code availability

All data and statistical analysis tools used in the present study are open source, details of which are available in the "Methods" section and Nature Portfolio Reporting Summary. No customized code was used to process or analyze data. PRS were calculated using PLINK 2.0 [https://www.cog-genomics.org/plink/2.0/] and weights for newly generated T2D PRS were estimated using PRS-CSx [https://github.com/getian107/PRScsx].

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

## Acknowledgements

This work was supported by the National Institutes of Health (R01HL151152 to C.K., C.G., and K.E.N., R00CA246063, R03CA287235, U01CA261339, R01HL174378, P50CA097186, P30CA015704, and U54HG013243 to B.F.D., R01HL143885 and R01DK139598 to P.G.-L. and K.E.N., R01HL163262, R01DK122503, and R01HL142302 to K.E.N., UM1DK078616 to C.K., R01HD30880 and R01AG065357 to P.G.-L., R01DK124097, R01HL156991, and R01DK123019 to R.J.F.L.), an award from the Andy Hill Cancer Research Endowment Distinguished Researchers Program (B.F.D.), and a Fred Hutchinson Cancer Center Translational Data Science New Collaboration Award (B.F.D). R.J.F.L. is supported by grants from the Novo Nordisk Foundation (NNF20OC0059313, NNF23SA0084103, and NNF18CC0034900). Work done by N.A.Y. was made possible by PPM2 grant #PPM2-0226–170020 from the Qatar National Research Fund (QNRF) and Qatar Genome Program (QGP) (members of Qatar Foundation) and by NPRP-S11 grant #NPRP11S-0114–180299 from the Qatar National Research Fund. The full acknowledgments and funding for the individual studies are listed in the Supplementary Information. The findings herein reflect the work and are solely the responsibility of the author. Funders did not contribute to the design or conduct of the study; collection, management, analysis, or interpretation of the data; or preparation, review, or approval of the manuscript.

## Author contributions

B.F.D., H.M.H., S.R., and R.A.J.S. contributed to study conception and design. B.G. and B.F.D. wrote the manuscript. C.K., C.G., and K.E.N. contributed to funding acquisition. Central data analysis group: B.G. (PAGE), D.K. (CCHC), Z.W. (BioMe), J.H. (AoU), R.T. (BioVU), K.A.B. (CLHNS), Y.W. (CHNS), N.P. and M.L. (CCPM), F.F.S. (Greenlandic), C.Y. (MESA), B.V. (MGI), K.R.I. and A.T.H. (MVP), N.A.Y. (Qatar), and A.D.P. (REGARDS) contributed to data analysis and interpretation. Study level principal investigators: K.L.M. (CLHNS), P.G.-L. (CHNS), A.A. (Greenlandic), M.B. (MGI), S.S.R. (MESA), A.M. and J.I.R. (MESA), R.M.I. (REGARDS), C.G. (CCPM), K.E.N. (CCHC), R.J.F.L. (BioMe), T.L.A. (MVP), C.A.H., L.L.M., and L.R.W. (MEC), and C.K. (WHI) contributed data and provided resource and administrative support. PAGE, AoU, BioMe, BioVU, CCHC, CCPM, CLHNS, CHNS, Greenlandic, MESA, MGI, MVP, Qatar, and REGARDS provided the data and administrative support. Additional contributions: Y.C., L.S., N.R., J. Shortt, L.W., M. Stanislawski, J. Pattee, L.D., P.S.S., M.M.S., N.J.C., N.R.L., M.E.J., P.B., C.L., T.H., I.M., J.B.M., D.O.S., X.Y., X.Z., K.-M.C., S.L.C., R.G.-S., J.L., P.S.T., S.B., M.G., L.M.R., Q.S., C.S.C., C.B.E., S.L., J.E.M., and U.P. contributed to design, data collection, management, coordination, interpretation, and/or analysis of contributing studies. B.F.D. and H.M.H. supervised the work. B.F.D. takes responsibility for the integrity and accuracy of the manuscript. All authors critically reviewed and approved the final version of the manuscript.

## Competing interests

R.J.F.L. has acted as a member of advisory boards and as a speaker for Ely Lilly and the Novo Nordisk Foundation, for which she has received fees. L.M.R. is a consultant for the NHLBI TOPMed Administrative Coordinating Center (through Westat). U.P. was a consultant with AbbVie, and her husband holds individual stocks for the following companies: BioNTech SE—ADR, Amazon, CureVac BV, Google/Alphabet Inc Class C, NVIDIA Corp, Microsoft Corp. The remaining authors declare no competing interests.

## Additional information

**Boya Guo** ®[1] ✉, **Yanwei Cai**[1], **Daeeun Kim** ®[2,3], **Roelof A. J. Smit**[4], **Zhe Wang** ®[4,5], **Kruthika R. Iyer**[6,7,8], **Austin T. Hilliard** ®[8], **Jeffrey Haessler**[1], **Ran Tao** ®[9,10], **K. Alaine Broadaway**[3], **Yujie Wang** ®[2], **Nikita Pozdeyev** ®[11], **Frederik F. Stæger** ®[12], **Chaojie Yang** ®[13], **Brett Vanderwerff**[14], **Amit D. Patki**[15], **Lauren Stalbow**[4], **Meng Lin**[11], **Nicholas Rafaels**[11],

Jonathan Shortt [11], Laura Wiley [11], Maggie Stanislawski [11], Jack Pattee[16], Lea Davis [17], Peter S. Straub[10,18], Megan M. Shuey [10,18], Nancy J. Cox[10,18], Nanette R. Lee[19], Marit E. Jørgensen [20,21], Peter Bjerregaard [21], Christina Larsen[21], Torben Hansen [22], Ida Moltke [12], James B. Meigs[23], Daniel O. Stram[24], Xianyong Yin [25,26,27], Xiang Zhou [26,27], Kyong-Mi Chang [28,29], Shoa L. Clarke [7,8,30], Rodrigo Guarischi-Sousa [7,8], Joanna Lankester[7,8], Philip S. Tsao [7,8], Steven Buyske [31], Mariaelisa Graff [2], Laura M. Raffield [3], Quan Sun [32], Lynne R. Wilkens[33], Christopher S. Carlson[1], Charles B. Easton[34,35], Simin Liu [36], JoAnn E. Manson [23], Loïc L. Marchand[33], Christopher A. Haiman [24], Karen L. Mohlke [3], Penny Gordon-Larsen [37], Anders Albrechtsen [12], Michael Boehnke [14], Stephen S. Rich [13], Ani Manichaikul [13], Jerome I. Rotter [38], Noha A. Yousri[39,40], Ryan M. Irvin[15], The biobank at the Colorado Center for Personalized Medicine (CCPM)*, VA Million Veteran Program (MVP)*, Chris Gignoux [11], Kari E. North [2], Ruth J. F. Loos [4], Themistocles L. Assimes [7,8], Ulrike Peters [1,41], Charles Kooperberg [1], Sridharan Raghavan[11,42], Heather M. Highland [2] & Burcu F. Darst [1] ✉

[1]Public Health Sciences, Fred Hutchinson Cancer Center, Seattle, WA, USA. [2]Department of Epidemiology, Gillings School of Global Public Health, University of North Carolina at Chapel Hill, Chapel Hill, NC, USA. [3]Department of Genetics, University of North Carolina at Chapel Hill, Chapel Hill, NC, USA. [4]The Charles Bronfman Institute for Personalized Medicine, Icahn School of Medicine at Mount Sinai, New York, NY, USA. [5]Department of Epidemiology, University of Alabama at Birmingham, Birmingham, AL, USA. [6]Data Science and Biotechnology, Gladstone Institutes, San Francisco, CA, USA. [7]Department of Medicine, Division of Cardiovascular Medicine, Stanford University School of Medicine, Stanford, CA, USA. [8]VA Palo Alto Health Care System, Palo Alto, CA, USA. [9]Department of Biostatistics, Vanderbilt University Medical Center, Nashville, TN, USA. [10]Vanderbilt Genetic Institute, Vanderbilt University of Medical Center, Nashville, TN, USA. [11]Colorado Center for Personalized Medicine, University of Colorado Anschutz Medical Campus, Aurora, CO, USA. [12]Department of Biology, Section for Computational and RNA Biology, University of Copenhagen, Copenhagen, Denmark. [13]Department of Genome Sciences, University of Virginia, Charlottesville, VA, USA. [14]Department of Biostatistics and Center for Statistical Genetics, University of Michigan, Ann Arbor, MI, USA. [15]Department of Epidemiology, School of Public Health, University of Alabama at Birmingham, Birmingham, AL, USA. [16]Department of Biostatistics and Informatics, Colorado School of Public Health, University of Colorado Anschutz Medical Campus, Aurora, CO, USA. [17]Department of Medicine, Division of Data-Driven and Digital Medicine, Icahn School of Medicine at Mount Sinai, New York, NY, USA. [18]Department of Medicine, Division of Genetic Medicine, Vanderbilt University of Medical Center, Nashville, TN, USA. [19]USC-Office of Population Studies Foundation, Inc., University of San Carlos, Cebu City, Philippines. [20]Steno Diabetes Center Greenland, Nuuk, Greenland. [21]Centre for Public Health in Greenland, National Institute of Public Health, University of Southern Denmark, Copenhagen, Denmark. [22]Novo Nordisk Foundation Center for Basic Metabolic Research, Faculty of Health and Medical Sciences, University of Copenhagen, Copenhagen, Denmark. [23]Department of Medicine, Mass General Brigham Healthcare System, Harvard Medical School, Boston, MA, USA. [24]Department of Preventive Medicine, Keck School of Medicine, University of Southern California, Los Angeles, CA, USA. [25]Department of Epidemiology, School of Public Health, Nanjing Medical University, Nanjing, China. [26]Department of Biostatistics, University of Michigan, Ann Arbor, MI, USA. [27]Center for Statistical Genetics, University of Michigan, Ann Arbor, MI, USA. [28]Corporal Michael J. Crescenz VA Medical Center, Philadelphia, PA, USA. [29]Department of Medicine, Division of Gastroenterology and Hepatology, University of Pennsylvania Perelman School of Medicine, Philadelphia, PA, USA. [30]Department of Medicine, Stanford Prevention Research Center, Stanford University School of Medicine, Stanford, CA, USA. [31]Department of Genetics, Rutgers University, Piscataway, NJ, USA. [32]Department of Biostatistics, Gillings School of Global Public Health, University of North Carolina at Chapel Hill, Chapel Hill, NC, USA. [33]Epidemiology Program, University of Hawaii Cancer Center, Honolulu, HI, USA. [34]Department of Epidemiology, School of Public Health, Brown University, Pawtucket, RI, USA. [35]Department of Family Medicine, Warren Alpert Medical School, Brown University, Pawtucket, RI, USA. [36]Department of Epidemiology, School of Public Health, Brown University, Providence, RI, USA. [37]Department of Nutrition, Gillings School of Global Public Health, University of North Carolina at Chapel Hill, Chapel Hill, NC, USA. [38]Department of Pediatrics, The Institute for Translational Genomics and Population Sciences, The Lundquist Institute for Biomedical Innovation at Harbor-UCLA Medical Center, Torrance, CA, USA. [39]College of Health and Life Sciences, Hamad Bin Khalifa University, Doha, Qatar. [40]Computer and Systems Engineering, Alexandria University, Alexandria, Egypt. [41]Department of Epidemiology, School of Public Health, University of Washington, Seattle, WA, USA. [42]Department of Medicine, Division of General Internal Medicine, University of Colorado School of Medicine, Denver, CO, USA. *Lists of authors and their affiliations appear at the end of the paper. Lists of members and their affiliations appears in the Supplementary Information. ✉e-mail: bguo@fredhutch.org; bdarst@fredhutch.org

## The biobank at the Colorado Center for Personalized Medicine (CCPM)

Heather D. Anderson[11], Christina L. Aquilante[11], Kelsey Arbogast[11], Christopher H. Arehart[11], Ian M. Brooks[11], Tonya M. Brunetti[11], Judith Brutus-Lestin[11], Elizabeth E. Burke[11], Emily M. Casteel[11], Joanne B. Cole[11], Curtis R. Coughlin II[11], Kristy Crooks[11], Jacob Crawford[11], Erin Culver[11], Michelle N. Edelmann[11], Matthew J. Fisher[11], Alan W. Franklin[11], Teresa C. Frye[11], Hunter George[11], Chris Gignoux[11], Elizabeth K. Gilliland[11], Casey S. Greene[11], Brooke Hawkes[11], Emily Hearst[11], Audrey E. Hendricks[11], Randi K. Johnson[11], Colleen G. Julian[11], Dave Kao[11], Iain Konigsberg[11], Lisa Ku[11], Elizabeth L. Kudron[11], Rashawnda Lacy[11], Ethan M. Lange[11], Yee Ming Lee[11], Joe A. Lesny[11], Meng Lin[11], Jan T. Lowery[11], Luciana B. Vargas[11], Betzaida L. Maldonado[11], Darcy Marceau[11], James L. Martin[11], Brianna L. Gates[11], David Mayer[11], Nicole L. McDaniel[11], Andrew Monte[11], Ethan Moore[11], Ann Nadrash[11], Jack Pattee[16], Nikita Pozdeyev [11], Alaa Radwan[11], Nick Rafaels[11], Sridharan Raghavan[11,42], Neda Rasouli[11], Elise L. Shalowitz[11], Hoda Sherif[11], Johnathan Shortt[11], Adrian M. Stewart[11], Kristen J. Sutton[11], Carolyn T. Swartz[11], Anna Tanaka[11], Matthew R. G. Taylor[11], Candace Teague[11], Emily B. Todd[11], Katy E. Trinkley[11] & Laura K. Wiley[11]

# VA Million Veteran Program (MVP)

Sumitra Muralidhar[43], Jennifer Moser[44], Jennifer E. Deen[44], Philip S. Tsao [7,8], J. Michael Gaziano[45], Elizabeth Hauser[46], Amy Kilbourne[47], Michael Matheny[48], Dave Oslin[49], Deepak Voora[46], Jessica V. Brewer[45], Mary T. Brophy[45], Kelly Cho[45], Lori Churby[8], Scott L. DuVall[50], Saiju Pyarajan[45], Robert Ringer[51], Luis E. Selva[45], Shahpoor Alex Shayan[45], Brady Stephens[52], Stacey B. Whitbourne[45], Themistocles L. Assimes [7,8], Adriana Hung[53] & Henry Kranzler[49]

[43]US Department of Veterans Affair, Washington, DC, USA. [44]US Department of Veterans Affairs, Washington, DC, USA. [45]VA Boston Healthcare System, Boston, MA, USA. [46]Durham VA Medical Center, Durham, NC, USA. [47]VA Ann Arbor Healthcare System, Ann Arbor, MI, USA. [48]VA Tennessee Valley Healthcare System, Murfreesboro, TN, USA. [49]Philadelphia VA Medical Center, Philadelphia, PA, USA. [50]VA Salt Lake City Health Care System, Salt Lake City, UT, USA. [51]New Mexico VA Health Care System, Albuquerque, NM, USA. [52]Canandaigua VA Medical Center, Canandaigua, NY, USA. [53]VA Tennessee Valley Healthcare System, Nashville, TN, USA.

