## [Transparent Peer Review file · Nature Communications]

Polygenic risk score for type 2 diabetes shows context-dependent effects across populations

Corresponding Author: Dr Boya Guo

Version 0:

Reviewer comments:

Reviewer #1

(Remarks to the Author)

This paper by Guo et al systematically T2D polygenic-risk-score (PRS) across several ancestries and contexts. I agree with the authors that there is value in taking a systematic approach like this to provide a single reference for how these scores behave, given the tendency for PRS to be dismissed out of hand if they are applied to ancestries other than the one on which they were trained.

The main weakness of the study is its novelty: almost all results confirm earlier work or are expected. For example, stronger genetic effects in younger adults and attenuation in hypertensive or obese groups is known. In fact, all of these could even be viewed as one finding, that ascertaining on lower environmental risk populations will produce lower PRS ORs. Most of the PheWAS hits are also well known based on analysis in other biobanks, and most of the new ones that emerge are also expected consequences of the metabolic syndrome.

My main suggestion is therefore to re-frame the introduction and discussion to more clearly state the impact of this study: providing a single reference of mostly known findings rather than finding fundamentally new clinical or biological things.

I would also make sure to cite previous work that has observed similar things to make it clear what is being replicated and what is novel.

I would emphasize the novelties more, such as the new ancestries that are analyzed.

The higher OR in males is worth a comment, as it is the main finding that doesn't follow the pattern of higher OR for lower environmental risk

I was not clear on why four scores needed to be tested when only one was used. Score 1 seems like the logical starting point. The authors should justify why they tested four scores and what was learned by doing so; alternatively, I would just remove the other three scores and keep things clean by doing score 1.

I would encourage the authors to release the full PheWAS results to a public catalog or web portal so that readers have full access to them.

Reviewer #2

(Remarks to the Author)

This manuscript reports on an extensive analysis of a multi-ancestry Type 2 Diabetes (T2D) polygenic risk score (PRS) across a large and diverse dataset encompassing 244,637 cases and 637,891 controls from eight populations in the PAGE Study and 13 additional biobanks/cohorts. The authors have characterized the performance of T2D PRS across different contexts and found notable dependencies on demographic and clinical factors, with better performance observed in younger individuals, males, and those without obesity or hypertension. The authors also demonstrate associations between the T2D PRS and various diabetes-related cardiometabolic traits and complications, suggesting potential clinical utility beyond basic risk prediction. The work highlights the importance of considering context when evaluating PRS as a prognostic tool and identifies generalizable associations with diabetes-related traits despite performance differences across diverse

populations.

Overall, this study provides valuable insights into the effects of T2D PRS on disease risk when stratified by demographic, medical, lifestyle, and behavioral factors. While many effects of these factors on T2D PRS have been previously reported in European populations, a significant strength of this work is the assessment across broad populations, revealing consistent results across ancestral groups - a novel finding to my knowledge. Additionally, the consistent effects of T2D PRS on various clinical comorbidities across populations represent an important contribution to the field.

Major Comments

1. On line 100, the authors state that PRS performance was better in those with a family history of T2D. However, Figure 3A shows no significant difference between those with and without family history of T2D in All, Hispanic, Asian, and European populations. The difference with $P = 0.04$ in African populations does not constitute a significant difference when considering multiple testing corrections. This discrepancy should be addressed.

2. Lines 367-371 should be rewritten to accurately reflect the findings regarding family history of T2D shown in Figure 3A, considering the comment above.

Minor Comments

1. In Figure 3A and Supplementary Figure 3A, "yes" and "no" appear in reverse order in the legends, which should be standardized for consistency.

2. In the captions of Figure 3, the authors state "P-values of heterogeneity are indicated when differences were statistically significant." Please clarify if this means P-values are shown only when less than 0.05. If so, this should be explicitly stated.

3. In Figure 4A, the T2D PRS was associated with increased log-transformed HOMA-IR in T2D controls only in European populations. This finding is compatible with epidemiological data and adds genetic evidence supporting the importance of insulin resistance in T2D pathophysiology in European populations.

4. Figure 4B rows 1 (SBP and DBP) and 2 (hypertension) should be reversed to maintain consistency with the order presented in the main text.

5. In Figure 4D, UACR and eGFR should be displayed on different scales, as the current presentation makes eGFR bar plots nearly unreadable.

6. Figure 5 is too visually dense. I suggest transposing the horizontal and vertical axes and enlarging the figures for better readability. Additionally, considering a log₁₀ scale for the y-axis (i.e., $-\log_{10}(P) = 1, 10, 100, 1000, \text{ and } 10000$ are plotted with the same intervals) may or may not improve visualization of the significance levels.

7. On line 309, please consider changing "S14" to "S4" for correct supplementary figure reference.

Reviewer #3

(Remarks to the Author)

Guo et al. manuscript is well written and presents a highly powered analysis aimed at improving our understanding of diabetes and polygenic risk score (PRS) prediction. The study offers valuable insights, emphasizing the context-specific nature of PRS effects. However, there is need of additional statistical tests to strengthen their conclusions and provide additional evidence to support key claims.

Major Comments:

1. Statistical Evidence for Context-Specific Effects

The central conclusion of the paper is that PRS effects are context-specific, is compelling but requires stronger statistical support. While differences in effect sizes across contexts are reported, the manuscript lacks formal interaction tests to demonstrate that these differences are statistically significant. I recommend including interaction terms (e.g., PRS × context) in regression models and testing their significance. Furthermore, replication of these interactions in an independent dataset would add robustness to the claim. This paper is well poised for this in view of it being well powered and leveraging a large array of biobank and cohort resources.

2. Differences in PRS Predictive Performance Across Contexts

In addition to examining effect sizes, the authors could significantly strengthen their main argument by evaluating whether the predictive accuracy (e.g., AUC, Naegelkerk R^2) of the PRS differs by context. Visual or tabular comparisons of PRS performance across subgroups would provide a more comprehensive view of context-dependency and offer translational insight.

3. Association with Age at Diagnosis

The authors have conducted an extensive PheWAS, which is commendable. An important and clinically relevant addition would be to assess whether the newly trained PRS is associated with age at diagnosis. Stratifying this analysis by PRS deciles would allow the authors to explore whether individuals at higher genetic risk tend to develop diabetes earlier in life - a finding that could enhance the clinical relevance of the PRS.

Version 1:

Reviewer comments:

Reviewer #1

(Remarks to the Author)

The authors have done a good job addressing my comments.

Reviewer #2

(Remarks to the Author)

The revised manuscript shows significant improvements and the authors have properly addressed all comments from myself and the other reviewers.

Minor Comments:

In the footnote for Figure 5, please correct the reference from "Table S12" to "Table S13".

The eGFR scale in Figure 5D appears problematic. Given that the unit is specified as ml/min/1.73m² in Table S13, a range of -0.05 to 0.05 seems unrealistically small. Please verify this scale and ensure it accurately represents the data.

I recommend a thorough review of all figures and tables to ensure consistency and prevent any discrepancies or mismatches.

Reviewer #3

(Remarks to the Author)

The authors have adequately addressed my major comments. However, one key aspect remains worth discussing. Although the AUC of the PRS is lower in African (AFR) and Hispanic (HIS) populations compared to those of European ancestry, the PRS still demonstrates utility. Specifically, it reveals that individuals with high genetic susceptibility in AFR and HIS groups tend to develop diabetes at an earlier age. This suggests that the PRS retains relevance in these populations and underscores the need for further studies to better understand why AFR and HIS individuals tend to experience earlier diabetes onset.

Response to Reviewers:

Dear Reviewers,

We would like to thank you for the valuable and constructive feedback provided on our manuscript. Please find below a point-by-point response to each reviewer's comments and a description of how we have addressed each comment. We believe the manuscript has been substantially strengthened by these suggestions. We appreciate the time and effort you invested in reviewing our work.

Reviewer #1:

This paper by Guo et al systematically T2D polygenic-risk-score (PRS) across several ancestries and contexts. I agree with the authors that there is value in taking a systematic approach like this to provide a single reference for how these scores behave, given the tendency for PRS to be dismissed out of hand if they are applied to ancestries other than the one on which they were trained.

- 1) The main weakness of the study is its novelty: almost all results confirm earlier work or are expected. For example, stronger genetic effects in younger adults and attenuation in hypertensive or obese groups is known. In fact, all of these could even be viewed as one finding, that ascertaining on lower environmental risk populations will produce lower PRS ORs. Most of the PheWAS hits are also well known based on analysis in other biobanks, and most of the new ones that emerge are also expected consequences of the metabolic syndrome.**

My main suggestion is therefore to re-frame the introduction and discussion to more clearly state the impact of this study: providing a single reference of mostly known findings rather than finding fundamentally new clinical or biological things. I would also make sure to cite previous work that has observed similar things to make it clear what is being replicated and what is novel. I would emphasize the novelties more, such as the new ancestries that are analyzed.

We thank Reviewer 1 for their comments and for recognizing the importance of clearly articulating the study's novelty and impact. We have revised the *Introduction* and *Discussion* to more explicitly indicate the novel contributions of this work, emphasizing that previous gene-environment T2D studies have primarily focused on individual genetic variants rather than PRS and were limited to European ancestry populations, while studies investigating PRS associations with diabetes-related complications focused on a limited number of outcomes, rather than a systematic investigation. We now frame this work as a comprehensive and systematic assessment of T2D PRS performance across diverse populations and contexts, which, to our knowledge, has not been previously conducted. The study objective, as revised in the *Introduction* now states that:

“The goal of this investigation is to improve precision medicine in diabetes by systematically evaluating the performance and context-dependent effects of a T2D PRS across diverse populations. Our findings serve as a comprehensive resource to characterize T2D PRS.”

We have also made revisions throughout the *Discussion* to better emphasize the significance and novel aspects of this research in the contexts of previous research. This includes modifying the first paragraph of the *Discussion*, as follows:

“In this large-scale T2D investigation of more than 240,000 T2D cases and 630,000 controls from PAGE and 13 independent biobanks and studies, we performed the first comprehensive and systematic cross-population characterization of a multi-ancestry genome-wide T2D PRS. This was achieved using a harmonized framework to evaluate context-dependent T2D PRS effects across different demographic and clinical characteristics, T2D PRS associations with diabetes-related traits, and a PheWAS of the T2D PRS. Our findings demonstrate that the PRS holds predictive value for T2D risk across diverse populations and was associated with younger age at T2D diagnosis. The T2D PRS demonstrated context-dependent associations with T2D risk, with PRS performance varying by certain demographic and clinical characteristics, such as age, sex, hypertension, and obesity status. Additionally, we found that genetic risk of T2D, as measured by the T2D PRS, was associated with other glycemic and cardiometabolic traits in controls, suggesting that PRS could provide insights into T2D etiology and serve as a valuable tool for predicting dysglycemia and metabolic risk, thereby facilitating early interventions and improving clinical risk assessment. Notably, associations between the T2D PRS and diabetes-related traits and complications were often consistent across populations, despite variable between-population PRS performance in estimating T2D risk, suggesting that the PRS may have clinical and prognostic utility beyond T2D risk prediction. Our PheWAS found associations between the T2D PRS and 40.3% of tested phenotypes, including phenotypes across all disease categories, underscoring the pleiotropic effects of T2D genetic risk and suggesting a broad biological impact of T2D-associated variants across organ systems. By providing a comprehensive resource that characterizes T2D PRS across diverse populations, our findings move beyond European-centric discovery and evaluate whether PRS utility—both predictive and prognostic—depends on individual- and population-level contexts. Future applications of T2D PRS in precision medicine should consider not only ancestry-specific calibration but also how environmental, metabolic, and clinical factors modulate the phenotypic expression of genetic risk.”

2) The higher OR in males is worth a comment, as it is the main finding that doesn't follow the pattern of higher OR for lower environmental risk

We agree that the higher odds ratio in males is an important and interesting finding. We have discussed this observation further in the *Discussion* section, dedicating a separate paragraph to explore potential explanations.

“Prior research on the performance of T2D PRS stratified by sex is limited^{9,30}, and evidence of sex-based differences in T2D heritability estimates is inconsistent^{31–34}. However, sex-based differences in PRS prediction have been widely reported across a range of complex traits.^{35–37} In this study, we observed stronger predictive performance of the T2D PRS in males. This difference could be attributed to several factors. First, certain genetic variants exhibit sex-dimorphic effects, with some loci demonstrating larger or more penetrant effects in males—for example, variants at *IRS1* show male-specific effects on fasting insulin and the 11p15.5 region, which harbors genes involved in insulin production such as *KCNQ1*, has been implicated in

male-specific T2D risk, potentially influenced by maternal inheritance patterns^{38–41}. Second, sex-specific biological mechanisms, including differences in glucose and lipid metabolism, affect fat distribution and insulin sensitivity, which can modulate the penetrance of these genetic variants^{40,41}. Additionally, diagnosis bias may lead to T2D being more frequently diagnosed in males⁴⁰. Future sex-specific GWAS and gene-by-sex interaction studies are needed to further investigate sex-specific genetic factors in T2D.”

3) I was not clear on why four scores needed to be tested when only one was used. Score 1 seems like the logical starting point. The authors should justify why they tested four scores and what was learned by doing so; alternatively, I would just remove the other three scores and keep things clean by doing score 1.

Identifying an optimal PRS for investigation was an important aspect of this work because our study includes unique populations where PRS performance has not been previously assessed. Further, PRS are rapidly superseded with emerging large-scale GWAS and novel methods. The PRS we ended up using (**PRS Method 1**, a previously developed PRS by Ge et al., 2022) was not clearly the best choice prior to our investigation as it was developed using T2D GWAS summary statistics that had smaller sample sizes than more recent T2D GWAS, and it did not include Hispanic participants (who are included in our investigation). As such, we evaluated four PRS, comparing different methods to construct PRS and different GWAS summary statistics. In PAGE, **PRS Method 1** had slightly better predictive performance across most populations compared to the other three PRS. Accordingly, **PRS Method 1** was used for all subsequent analyses.

To better motivate this component of our investigation, we now clarify in the *Results – Evaluation of T2D PRS performance* section that:

“Given our unique populations and how rapidly PRS are superseded, our first goal was to identify a T2D PRS that had optimal predictive ability across our populations to be carried forward in subsequent PRS evaluations.”

The subsequent text of this section has also been modified to clarify how the four PRS differ from one another and further motivate this aspect of the study, as follows:

“In PAGE, we assessed four distinct multi-ancestry PRS (**Figure 1, Table S3, Methods**). The first three PRS were genome-wide PRS developed using 1.259 million HapMap3 variants in PRS-CSx¹², an approach that has demonstrated high predictive ability across several traits^{13–15}. **PRS Method 1**: a previously developed T2D PRS¹⁶ based on AFR¹⁷, ASN¹⁸, and EUR¹⁹ GWAS summary statistics using meta-analyzed posterior effects with the META=T function in PRS-CSx; **PRS Method 2**: a new T2D PRS that we developed using a similar approach as PRS Method 1 but leveraging GWAS summary statistics from larger sample sizes of AFR, ASN, and EUR populations and also including HIS and South Asian (SAS) populations^{6,7}; **PRS Method 3**: a new T2D PRS that we developed using the same summary statistics as PRS Method 2 but calculated from a linear combination of population-specific PRS using the META=F function in PRS-CSx; and **PRS Method 4**: a previously developed PRS with 582 genome-wide significant

variants reported in AFR, ASN, EUR, and HIS populations^{7,8}. PRS derived from each method were standardized to the distribution of controls within each population.”

Further, we have modified **Table S3** to facilitate comparisons of the four PRS, particularly the methods, studies, and population sample sizes used to construct each PRS.

4) I would encourage the authors to release the full PheWAS results to a public catalog or web portal so that readers have full access to them.

As suggested, we have made the full meta-analyzed PheWAS results, along with the study-specific PheWAS results from each of the five contributing biobanks (BioMe, BioVU, CCPM, MVP, and MGI), publicly available on Zenodo, which can be accessed at the following link: <https://doi.org/10.5281/zenodo.15998801>. This is now described in the Data availability statement. The meta-analyzed PheWAS results for the T2D PRS—both across all populations and stratified by population—are also presented in **Table S16**.

Reviewer #2:

This manuscript reports on an extensive analysis of a multi-ancestry Type 2 Diabetes (T2D) polygenic risk score (PRS) across a large and diverse dataset encompassing 244,637 cases and 637,891 controls from eight populations in the PAGE Study and 13 additional biobanks/cohorts. The authors have characterized the performance of T2D PRS across different contexts and found notable dependencies on demographic and clinical factors, with better performance observed in younger individuals, males, and those without obesity or hypertension. The authors also demonstrate associations between the T2D PRS and various diabetes-related cardiometabolic traits and complications, suggesting potential clinical utility beyond basic risk prediction. The work highlights the importance of considering context when evaluating PRS as a prognostic tool and identifies generalizable associations with diabetes-related traits despite performance differences across diverse populations.

Overall, this study provides valuable insights into the effects of T2D PRS on disease risk when stratified by demographic, medical, lifestyle, and behavioral factors. While many effects of these factors on T2D PRS have been previously reported in European populations, a significant strength of this work is the assessment across broad populations, revealing consistent results across ancestral groups - a novel finding to my knowledge. Additionally, the consistent effects of T2D PRS on various clinical comorbidities across populations represent an important contribution to the field.

Major Comments

1. On line 100, the authors state that PRS performance was better in those with a family history of T2D. However, Figure 3A shows no significant difference between those with and without family history of T2D in All, Hispanic, Asian, and European populations. The difference with $P = 0.04$ in African populations does not constitute a significant difference when considering multiple testing corrections. This discrepancy should be addressed.

We thank the reviewer for pointing this out. In response to the reviewer's suggestion, we have removed the mention of differences in PRS effects by family history of T2D from the abstract, *Results – Context-dependent effects of T2D PRS* section, and the *Discussion* since the evidence was not consistent across meta-analyses.

2. Lines 367-371 should be rewritten to accurately reflect the findings regarding family history of T2D shown in Figure 3A, considering the comment above.

As suggested, we have removed the discussion of PRS effects by family history of T2D.

Minor Comments

1. In Figure 3A and Supplementary Figure 3A, "yes" and "no" appear in reverse order in the legends, which should be standardized for consistency.

As suggested, we changed the order of “Yes” and “No” in the **Figure S3** to match the order in **Figure 3** (now **Figure 4**).

2. In the captions of Figure 3, the authors state "P-values of heterogeneity are indicated when differences were statistically significant." Please clarify if this means P-values are shown only when less than 0.05. If so, this should be explicitly stated.

In response to this comment, we have revised the captions for **Figure 3** (now **Figure 4**) and **Figure S3** to explicitly indicate that P-values are shown only when statistically significant ($P < 0.05$). The significance threshold for stratified analyses is also stated in the *Methods* section:

“A $P < 0.05$ threshold was used to determine statistical significance for stratification analyses.”

3. In Figure 4A, the T2D PRS was associated with increased log-transformed HOMA-IR in T2D controls only in European populations. This finding is compatible with epidemiological data and adds genetic evidence supporting the importance of insulin resistance in T2D pathophysiology in European populations.

We thank the reviewer for this observation. We have added the following sentences to the *Discussion* to highlight this finding:

“Our findings also have important implications for T2D etiology and pathophysiology. For instance, the positive association between T2D PRS and HOMA-IR in EUR controls provides genetic evidence that complements previous epidemiological studies on the critical role of insulin resistance in T2D development^{42–44}. This association suggests that genetic factors contribute to insulin resistance pathways even before clinical diabetes onset, offering mechanistic insights into how genetic predisposition translates into disease risk.”

4. Figure 4B rows 1 (SBP and DBP) and 2 (hypertension) should be reversed to maintain consistency with the order presented in the main text.

As suggested, we have updated **Figure 4B** (now **Figure 5B**) by moving hypertension above SBP and DBP to maintain consistency with the order presented in the main text.

5. In Figure 4D, UACR and eGFR should be displayed on different scales, as the current presentation makes eGFR bar plots nearly unreadable.

As suggested, we updated **Figure 4D** (now **Figure 5D**) so that UACR and eGFR are now displayed on different scales.

6. Figure 5 is too visually dense. I suggest transposing the horizontal and vertical axes and enlarging the figures for better readability. Additionally, considering a log₁₀ scale for the y-axis (i.e., -log₁₀(P) = 1, 10, 100, 1000, and 10000 are plotted with the same intervals) may or may not improve visualization of the significance levels.

We thank the reviewer for this suggestion. To improve readability, we enlarged the figure (now **Figure 6**) and removed the inset plot, as PheWAS results are also presented in **Table 2** and **Table S17** (and now also in Zenodo). The Y-axis is plotted using the -log₁₀(P) scale. To improve visualization, we applied a square root transformation to the Y-axis. This compresses extremely small P-values while improving visualization for modest signals without changing the Y-axis label. Since we have implemented these changes to improve the visualization of this figure and because PheWAS results are typically presented using Manhattan plots, we used the same orientation of the X- and Y-axes.

7. On line 309, please consider changing "S14" to "S4" for correct supplementary figure reference.

Thank you for pointing that out. We have corrected the supplementary figure reference from "S14" to "S4" in the sentence referenced by the reviewer.

Reviewer #3:

Guo et al. manuscript is well written and presents a highly powered analysis aimed at improving our understanding of diabetes and polygenic risk score (PRS) prediction. The study offers valuable insights, emphasizing the context-specific nature of PRS effects. However, there is need of additional statistical tests to strengthen their conclusions and provide additional evidence to support key claims.

Major Comments:

1. Statistical Evidence for Context-Specific Effects

The central conclusion of the paper is that PRS effects are context-specific, is compelling but requires stronger statistical support. While differences in effect sizes across contexts are reported, the manuscript lacks formal interaction tests to demonstrate that these differences are statistically significant. I recommend including interaction terms (e.g., PRS × context) in regression models and testing their significance. Furthermore, replication of these interactions in an independent dataset would add robustness to the claim. This paper is well poised for this in view of it be well powered and leveraging a large array of biobank and cohort resources.

We thank the reviewer for this comment. Our interpretation of PRS having context-dependent effects is based on statistical evidence of effect heterogeneity between strata, evaluated using Cochran's Q-test—commonly used in meta-analyses to assess whether variation in effect sizes across subgroups or studies exceeds what would be expected by chance. These heterogeneity P-values were previously shown in **Figure 4** and **Figure S3**, and we have now also added these results to **Table S9** and **Table S11** to present all heterogeneity evidence.

Further, as suggested, in PAGE we have performed interaction tests between the T2D PRS and each context factor, presented in a new table (**Table S12**). We report the odds ratios and corresponding P-values for each interaction term, using the lowest-risk context group as the reference. In addition, we provide likelihood ratio test P-values comparing models with and without the interaction term to assess the overall significance of including the interaction term. Results from tests of interaction align with those from Cochran's Q-test, further supporting the presence of context-specific differences in PRS effects.

These statistical tests of context-dependent effects are now described in the first sentence of the *Results – Context-dependent effects of T2D PRS* section, as follows:

“We investigated whether the T2D PRS demonstrated context-dependent effects on T2D risk by conducting association analyses stratified by 18 demographic, medical, and lifestyle and behavioral characteristics and evaluating heterogeneity using Cochran's Q-test and interaction terms in regression models (**Methods and Table S7**).”

The interaction analyses are now described in the *Methods – Stratification analyses* section as follows:

“We further evaluated evidence of context-dependent effects of PRS within PAGE by including an interaction term between the PRS and each context variable, also adjusting for the main effects of each. A likelihood ratio test was used to compare models with and without the interaction term.”

Last, we agree that it is important to replicate interactions in independent datasets to ensure that findings are robust. While we present our primary results as those meta-analyzed between PAGE (N=82,944) and the 13 independent biobanks and cohorts (N=835,241), as this maximizes power for discovery, all context-dependent associations (**Figure S3, Tables S10 and S11**) and associations with diabetes-related traits (**Tables S15 and S16**) are presented separately for

PAGE and the 13 biobanks and cohorts for the purpose of providing replication. We have included a supplemental table (**Table S8**) that describes context-dependent findings in PAGE, the 13 biobanks and cohorts, and meta-analyses across all studies to summarize what findings have been replicated across studies and populations.

This strength of having multiple lines of evidence for replication is now described as follows in the *Results – Context-dependent effects of T2D PRS* section, as follows:

“In analyses performed separately in PAGE versus the additional biobanks and studies, evidence of replication was observed for T2D PRS effects when stratifying by age groups, sex, hypertension and obesity status (**Figure S3, Tables S10 and S11**), as summarized in **Table S8**.”

Likewise, this is also emphasized in the *Results – Association between T2D PRS and diabetes-related traits* section, as follows:

“Significant associations (applying a $P < 2.50 \times 10^{-3}$ [0.05/20 traits] Bonferroni adjusted threshold) were found across various glycemic and cardiometabolic traits, including HbA1c, fasting glucose, hypertension, systolic blood pressure (SBP), diastolic blood pressure (DBP), triglycerides, non-HDL cholesterol, and HDL cholesterol (**Figure 5, Table S14**), with consistent results observed in analyses performed separately in PAGE versus the additional biobanks and studies (**Tables S15 and S16**).”

2. Differences in PRS Predictive Performance Across Contexts

In addition to examining effect sizes, the authors could significantly strengthen their main argument by evaluating whether the predictive accuracy (e.g., AUC, Naegelkerk R²) of the PRS differs by context. Visual or tabular comparisons of PRS performance across subgroups would provide a more comprehensive view of context-dependency and offer translational insight.

AUCs for analyses stratified by contexts—for models including and excluding PRS and models of PRS alone—are reported separately for PAGE and for the additional biobanks and cohorts, as now referenced in the *Results – Context-dependent effects of T2D PRS* section, as follows:

“AUCs calculated separately in each stratum further supported these findings (**Table S10**).”

In addition, we reported both AUC (for binary traits) and R² (for continuous traits) for the association between the T2D PRS and diabetes-related traits, separately for PAGE and the additional biobanks and cohorts, as now referenced in the *Results – Association between T2D PRS and diabetes-related traits*, as follows:

“AUC and R² estimates further supported these findings (**Table S15**).”

3. Association with Age at Diagnosis

The authors have conducted an extensive PheWAS, which is commendable. An important and clinically relevant addition would be to assess whether the newly trained PRS is associated with age at diagnosis. Stratifying this analysis by PRS deciles would allow the

authors to explore whether individuals at higher genetic risk tend to develop diabetes earlier in life a finding that could enhance the clinical relevance of the PRS.

We appreciate the reviewer’s suggestion to examine the association between the T2D PRS and age at diagnosis, which is indeed a clinically relevant aspect of T2D risk stratification. In response, we plotted the distribution of mean age at diagnosis by T2D PRS decile and population in All of Us (now included as **Figure 3**) and formally tested the association between PRS and age at diagnosis (**Table S6**). Overall, we found that higher T2D PRS was significantly associated with younger age at diagnosis in each population and across all populations combined.

This analysis is now described in a new *Methods – Association between T2D PRS and age at T2D diagnosis* section, as follows:

“We evaluated the association between the T2D PRS and age at T2D diagnosis in a case-only analysis in All of Us. This analysis was limited to All of Us due to its large sample size, inclusion of multiple populations, and the ability to estimate age at diagnosis. Age at diagnosis was determined by the earliest occurrence start date of T2D mellitus (SNOMED concept ID: 201826), T2D mellitus without complication (SNOMED concept ID: 4193704), or disorder due to T2D mellitus (SNOMED concept ID: 443732). We tested the association between each PRS category (predictor) and age at diagnosis (outcome) in linear regression models, using the 0–10% PRS decile as the reference. Analyses were performed separately in each population and across all populations.”

We also added the following to the *Results – Evaluation of T2D PRS performance* section to report these findings:

“As a secondary analysis, we evaluated the association between the T2D PRS and age at T2D diagnosis (**Figure 3 and Table S6**). Higher T2D PRS was significantly associated with younger age at diagnosis across populations. Individuals with T2D in the top 10% PRS decile were diagnosed, on average, 8.9 years earlier (95% CI: -9.83 to -7.89 ; $P=1.09\times 10^{-78}$) than those in the bottom 10% (**Table S6**). This pattern was consistent across populations, with individuals in the top 10% PRS decile diagnosed on average 3.79 years earlier in AFR ($P=1.48\times 10^{-9}$), 10.17 years earlier in ASN ($P=1.67\times 10^{-3}$), 2.46 years earlier in EUR ($P=3.33\times 10^{-4}$), and 4.86 years earlier in HIS ($P=8.75\times 10^{-7}$) populations compared to those in the bottom 10% decile.”

This finding is also now briefly mentioned in the first paragraph of the *Discussion* section:

“Our findings demonstrate that the PRS holds predictive value for T2D risk across diverse populations and was associated with younger age at T2D diagnosis.”

Response to Reviewers:

Dear Reviewers,

We would like to thank you for your continued careful review and valuable suggestions on our manuscript. We have addressed all remaining points as detailed below, and we believe these final changes further improve the clarity and quality of the work.

Reviewer #1:

The authors have done a good job addressing my comments.

Reviewer #2:

The revised manuscript shows significant improvements and the authors have properly addressed all comments from myself and the other reviewers.

Minor Comments:

- 1. In the footnote for Figure 5, please correct the reference from "Table S12" to "Table S13".**

Thank you for pointing that out. As suggested, we have corrected the reference from Table S12 to Table S13.

- 2. The eGFR scale in Figure 5D appears problematic. Given that the unit is specified as ml/min/1.73m² in Table S13, a range of -0.05 to 0.05 seems unrealistically small. Please verify this scale and ensure it accurately represents the data.**

We appreciate the opportunity to clarify this point. In Figure 5D (now Supplementary Figure 4B, as the former Figure 5C/D/E are now in a new Supplementary Figure 4 to follow *Nature Communications* figure size requirements), the x-axis represents beta estimates and 95% confidence intervals for associations between PRS and eGFR. Among all T2D cases, each SD increase in T2D PRS was associated with a -0.03 mL/min/1.73 m² (95% CI: -0.04 , -0.02) lower eGFR, and in EUR T2D cases, each SD increase in T2D PRS was associated with a -0.04 mL/min/1.73 m² (95% CI: -0.06 , -0.03) lower eGFR.

The magnitude of these associations is fairly similar to those reported in a previous study (Udler et al., *PLoS Medicine* 2018) that found a significant inverse association between a liver/lipid PRS and eGFR ($\beta = -0.002$ mL/min/1.73 m², $P = 1.2 \times 10^{-6}$).

- 3. I recommend a thorough review of all figures and tables to ensure consistency and prevent any discrepancies or mismatches.**

Thank you for this suggestion. We have carefully reviewed all figures and tables to ensure their accuracy and consistency and formatted the manuscript in accordance with the submission guidelines of *Nature Communications*.

Reviewer #3

The authors have adequately addressed my major comments.

- 1. However, one key aspect remains worth discussing. Although the AUC of the PRS is lower in African (AFR) and Hispanic (HIS) populations compared to those of European ancestry, the PRS still demonstrates utility. Specifically, it reveals that individuals with high genetic susceptibility in AFR and HIS groups tend to develop diabetes at an earlier age. This suggests that the PRS retains relevance in these populations and underscores the need for further studies to better understand why AFR and HIS individuals tend to experience earlier diabetes onset.**

We thank the reviewer for this comment and agree this is an important point to discuss. As suggested, we've added the paragraph below to the *Discussion* section:

“Despite the lower predictive ability of the T2D PRS in AFR and HIS populations, we observed the strongest evidence of the PRS being associated with a younger age at T2D diagnosis in these two populations, demonstrating potential clinical utility. AFR and HIS populations tended to develop diabetes earlier than ASN and EUR populations; while these differences could be partially attributed to the younger age at enrollment in these populations (mean age at enrollment: AFR = 54.7 [SD: 10.2], HIS = 53.3 [SD: 11.4], ASN = 54.5 [SD: 12.5], EUR = 60.5 [SD: 12.5]), similar differences in age at T2D diagnosis have been previously reported. As such, these differences may reflect complex interactions between genetic and non-genetic factors that may warrant further studies to understand the causes and mechanisms of earlier T2D onset in these groups.”